# TsLLM: Augmenting LLMs for General Time Series Understanding and Prediction

Felix Parker [1]   Nimeesha Chan [1]   Chi Zhang [1 2]   Kimia Ghobadi [1]

## Abstract

Time series data is fundamental to decision-making across many domains including healthcare, finance, power systems, and logistics. However, analyzing this data correctly often requires incorporating unstructured contextual information, answering domain-specific questions, and generating natural language explanations – capabilities that traditional time series models lack. While Large Language Models (LLMs) excel at contextual reasoning and knowledge integration, they struggle with numerical time series due to inefficient text-based representations and limited exposure to numerical data during pretraining. We address this gap by augmenting an LLM with specialized time series perception through a patch-based encoder-decoder architecture. We train this time series-augmented LLM (TsLLM) on a large corpus of over 25 billion tokens of interleaved time series and text spanning diverse tasks: forecasting with contextual information, question-answering, anomaly detection, classification, report generation, and more, all unified as autoregressive next token prediction. This training enables TsLLM to leverage both its natural language skills and newly acquired understanding of numerical time series signals. While not designed to surpass specialized models on traditional benchmarks, TsLLM demonstrates strong performance on tasks requiring the integration of time series analysis with natural language – capabilities that conventional approaches cannot provide. It also exhibits strong zero-shot and few-shot performance, showing it can adapt to new data without additional training.

[1]Center for Systems Science and Engineering, Johns Hopkins University, Baltimore, MD, USA [2]Department of Computer Science, Johns Hopkins University, Baltimore, MD, USA. Correspondence to: Felix Parker <fparker9@jhu.edu>.

*Proceedings of the 43rd International Conference on Machine Learning*, Seoul, South Korea. PMLR 306, 2026. Copyright 2026 by the author(s).

## 1. Introduction

Traditional time series models are effective for narrowly defined numerical tasks, mapping structured inputs to structured outputs; however, many real-world applications of time series analysis require more than numerical prediction – context is often crucial. Clinicians use electrocardiograms for diagnosis in the context of symptoms and lab results, analysts forecast while conditioning on news, and power grid operators forecast demand and manage supply using load, frequency, and generation traces together with weather forecasts, outage reports, and event calendars. Across healthcare, finance, scientific research, and many other fields, domain experts need systems that can understand both numerical time series data and all relevant contextual information, draw on domain knowledge, make predictions, and effectively communicate findings.

Large language models (LLMs) possess the capabilities needed for these integrated tasks: contextual reasoning, domain knowledge integration, few-shot learning from examples, and natural language interaction. As a result, utilizing LLMs to perform certain time series analysis tasks holds considerable promise (Jin et al., 2024a; Zhang et al., 2024). However, their weak abilities to perceive numerical time series data prevent LLMs from effectively applying their capabilities to temporal analysis (Tan et al., 2024; Merrill et al., 2024). Two structural issues explain this gap. First, text tokenization represents each numeric value with multiple tokens, awkwardly splitting up points and inflating sequence lengths dramatically; this limits the amount of signal and context that fits in context windows and hinders pattern recognition. A single 10-second electrocardiogram used in our experiments (12 leads, 100 Hz) requires 84,000 tokens using a standard tokenizer – longer than the maximum context length of many models. Second, our analysis of LLM pretraining corpora reveals they contain little high-quality time series data, so LLMs lack the exposure needed for understanding and analysis.

To address these limitations, we introduce TsLLM, a large language model augmented with dedicated time series perception that enables unified, interleaved text-time series reasoning and generation. TsLLM adds a high-fidelity, patch-based encoder-decoder that is pretrained with a $\beta$-

VAE objective to produce compact, information-dense representations of temporal signals, along with a scale-aware encoding that decouples shape from magnitude so the model can reason about absolute scale while maintaining numerical stability. A lightweight cross-attention-based adapter then maps these time series representations into the LLM's embedding space to bridge the domain gap. TsLLM's inputs and outputs can interleave text and time series segments, and the model treats all tasks as next-token prediction, unifying these modalities. This preserves the pretrained LLM's language ability while adding precise temporal perception.

TsLLM's unified formulation of language and time series modeling enables functions that traditional time series models and text-only LLMs generally do not provide: contextual forecasting conditioned on unstructured information, question answering over numerical time series data, and natural language guided time series synthesis. It also supports standard time series tasks, including forecasting, classification, anomaly detection and more, without using task-specific heads. Few-shot in-context prompting with interleaved examples further allows the model to adapt to new datasets and tasks without additional fine-tuning.

Empirically, TsLLM demonstrates strong performance on tasks that require integrating temporal signals with language. TsLLM significantly outperforms frontier LLMs on time series question-answering (QA) benchmarks, matches or exceeds specialized systems for QA on electrocardiogram data, and beats both LLM-based methods and traditional models on a contextual forecasting benchmark. On traditional time series benchmarks, TsLLM is competitive with dedicated models on forecasting and classification, even though it is not optimized for purely numerical tasks. It does this while maintaining the performance of its base model on standard text-only LLM evaluations. Ablations show that few-shot prompting provides significant gains in most settings, and that the performance of TsLLM scales with the size of the pretrained LLM it is built on, suggesting that its performance can be improved by scaling up further.

The primary contributions of this work are:

1. The TsLLM model architecture, which integrates a time series encoder and decoder with a pretrained LLM, and allows for interleaved text and time series inputs and outputs.
2. A high-fidelity patch-based encoder-decoder for time series signals that is pretrained on a diverse set of time series data with a $\beta$-VAE objective to produce compact, information-dense representations for use in the LLM.
3. A multi-stage training procedure that aligns the LLM with the encoder and decoder models, then trains it to solve a wide range of time series analysis tasks on diverse data, adding temporal perception abilities to the LLM while preserving its language competence.

4. Strong empirical results on benchmarks that require integrating temporal signals with language, as well as traditional time series tasks.
5. A large corpus of multimodal interleaved text and numerical time series sequences, comprising over 25 billion tokens in total, which were collected, augmented, and synthesized for training TsLLM.
6. The full code and data for TsLLM are available at `https://github.com/flixpar/TsLLM` to enable further research in this area.

Our goal is not to replace specialized time series models on pure numerical tasks, as they are effective and efficient for many time series-specific tasks. Rather, TsLLM targets problems that require integrating temporal analysis with domain knowledge and natural-language interaction. We view TsLLM as complementary to dedicated time series models and foundation models: it expands what practitioners can do with temporal data by enabling direct, language-based interaction that is grounded in the underlying signals.

## 2. Related Work

**Time series foundation models and numerical temporal modeling.** Time series foundation models (TSFMs) pretrain on large collections of numerical sequences to learn reusable temporal representations for forecasting and related structured prediction tasks (Das et al., 2024; Goswami et al., 2024; Liu et al., 2025a). These models have substantially improved generalization across domains and horizons, and some systems, such as MOMENT, extend beyond single-task forecasting through unified task specifications or lightweight task-specific decoders (Goswami et al., 2024). However, TSFMs generally operate over numerical inputs and numerical outputs, with limited ability to incorporate unstructured textual context, answer domain-specific natural language questions, or produce explanations. This makes them well suited for pure temporal prediction, but less suited for applications in which a time series must be interpreted jointly with clinical notes, news, metadata, reports, or user instructions. TsLLM is designed to complement rather than replace these models: it targets the setting where temporal perception must be combined with the contextual reasoning and interactive capabilities of LLMs.

**LLMs for time series prediction.** A parallel line of work adapts pretrained language models for time series forecasting. Early approaches showed that LLMs could be prompted with serialized numerical sequences for zero-shot forecasting (Gruver et al., 2023; Xue and Salim, 2024), while subsequent work improved efficiency and accuracy through reprogramming, patching, multiscale decomposition, quantization, or vocabulary expansion (Zhou et al., 2023; Jin et al., 2024b; Ansari et al., 2024; Chan et al., 2024; Kowsher et al.,

2025; Zhang et al., 2026). These methods demonstrated that pretrained LLMs can provide useful sequential inductive bias, but many use the LLM primarily as a sequence model rather than as a language model: textual context is absent, weakly used, or restricted to narrow prompts, and the outputs are usually forecasts rather than arbitrary mixtures of text and time series. They also often train only lightweight adapters or external encoders, leaving the LLM itself only partially aligned with temporal data.

**Time series understanding with modern LLMs and MLLMs.** Several studies have evaluated whether LLMs can directly understand and reason about time series. Early 2024 work found that text-only LLMs struggled with numerical temporal perception, feature recognition, and causal or etiological reasoning (Tan et al., 2024; Merrill et al., 2024; Fons et al., 2024). While LLMs have improved dramatically since then, particularly in reasoning and multimodal capabilities, the core representation problem identified by this work remains relevant. Raw time series are poorly matched to text tokenizers, long numerical sequences consume large context windows, and text pretraining provides limited exposure to high-quality temporal signals. Modern reasoning models and vision-language models can partially compensate through better inference and chart understanding, but they still lack a native, efficient, and high-fidelity interface to dense numerical time series. This motivates augmenting LLMs with specialized temporal perception rather than relying solely on scale or prompting.

**Multimodal and language-integrated time series models.** Recent work has explored several ways to provide LLMs with temporal inputs. Visualization-based methods convert time series into plots and use vision-language models for reasoning (Daswani et al., 2024; Liu et al., 2025b). This strategy has become increasingly viable as multimodal LLMs have improved, and it offers a simple way to expose temporal shape without modifying the model. However, plots can lose information about exact values, sampling rate, scale, dense multivariate structure, and high-frequency dynamics; they also do not naturally support generation of raw time series outputs. Other approaches introduce explicit time series tokens or learned adapters. ChatTime quantizes continuous values into discrete tokens (Wang et al., 2025), ChatTS maps time series patches into the LLM embedding space using synthetic instruction data (Xie et al., 2025), and Chat-TS adds time series tokens to an LLM vocabulary for multimodal reasoning over text and temporal data (Quinlan et al., 2026). Domain-specific systems such as OpenTSLM focus on medical time-series-language reasoning (Langer et al., 2025), while TSLA develops a multi-task time series language model with joint time-series and language modeling (Fang et al., 2025). These models are close in spirit to TsLLM: they share the view that LLMs need specialized

temporal interfaces and training data to reason over time series. TsLLM differs in combining a continuous, high-fidelity, scale-aware patch encoder-decoder with arbitrary interleaving of text and time series in both inputs and outputs, enabling not only time-series-conditioned language generation but also language-guided time series generation.

## 3. Methodology

We propose TsLLM, a multimodal generative model that augments a pretrained decoder-only LLM with time series perception and generation abilities. Our design goals are: (a) build good representations of time series signals that are structured and compact without significant information loss; (b) allow for a mixture of text and time series in inputs and outputs to provide flexibility in constructing tasks; and (c) ensure the model has sufficient training to allow it to perform well across a highly diverse set of tasks and data. To meet these goals, TsLLM consists of four main components: (1) a patch-based time series encoder-decoder, (2) a pretrained LLM, (3) a pair of fusion layers that inject time series embeddings into the LLM token stream and map LLM hidden states back to time series patches, and (4) a multi-stage training pipeline. These components are illustrated in Figure 1.

Our unified framework reformulates all time series tasks, including forecasting, classification, anomaly detection, and general question-answering, as autoregressive generation problems over interleaved sequences of text and time series tokens. This approach enables few-shot generalization to new domains through natural language instructions and in-context examples, while supporting rich contextual information such as domain knowledge, metadata, and even structured tabular data. The following sections detail each component that are primary contributions of this work, namely, our time series encoding mechanism (Sec. 3.1), the architectural modifications to the LLM (Sec. 3.2), and our training strategy (Sec. 3.3).

### 3.1. Time Series Encoding and Decoding

TsLLM depends on high-quality representations of time series signals for strong performance. However, current LLMs split individual numbers into multiple tokens, making it hard for models to capture numerical relationships and resulting in extremely long, inefficient inputs, with each time point using four to eight tokens (including delimiters) (Spathis and Kawsar, 2024). This limits the amount of time series and contextual information, particularly covariates and few-shot examples, that can fit in the context window, and makes training very computationally expensive. An alternative time series encoding is needed.

We solve this representation challenge through a patch-

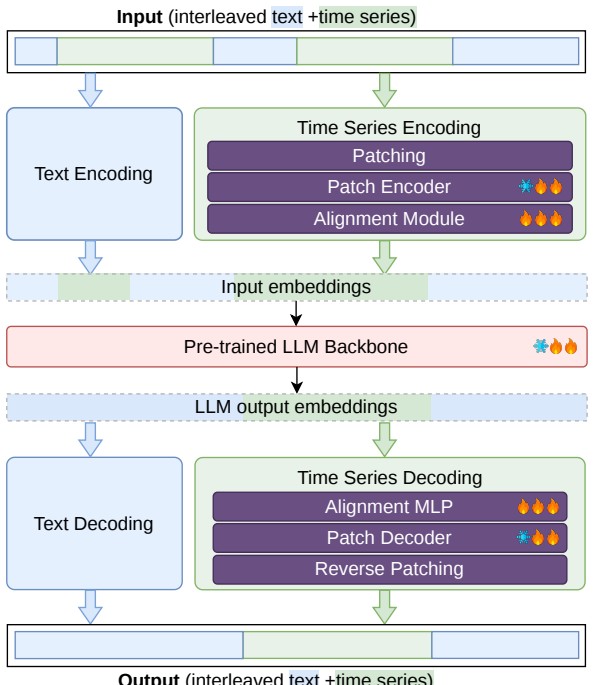

*Figure 1.* The TsLLM architecture, with interleaving of text (blue) and time series (green) inputs and outputs. The time series encoder-decoder has been pretrained to produce high-quality time series patch representations, which our multi-stage alignment process (❄ = frozen, 🔥 = trained) bridges with the LLM to enable unified text and time series processing.

based encoding mechanism that preserves information while achieving dramatic compression. Our approach segments time series into fixed-length patches of $p$ consecutive points, and transforms each patch $\mathbf{x}_{i:i+p}$ into a single dense representation through a learned MLP: $\mathbf{z}_i^{\text{local}} = \text{MLP}_{\text{patch}}(\mathbf{x}_{i:i+p})$. The patch size $p = 32$ was selected through extensive ablation studies (Table 5), providing an optimal trade-off: smaller patches preserve more detail but increase sequence length, while larger patches risk losing important local features. Each component in a multivariate signal is encoded independently, allowing for flexibility in the number of components. The resulting univariate signals are fed into the LLM sequentially, allowing it to learn the relationships itself.

Real-world time series exhibit enormous variability in scale – from microvolts in neural recordings to millions in financial transactions. To ensure stable training across this range, we apply sequence-level z-score normalization before encoding: $\hat{\mathbf{x}} = (\mathbf{x} - \mu_{\mathbf{x}})/\sigma_{\mathbf{x}}$, where $\mu_{\mathbf{x}}$ and $\sigma_{\mathbf{x}}$ are computed over the entire input sequence.

We pretrain the encoder-decoder pair using a $\beta$-VAE objective that balances reconstruction fidelity with representation quality. The loss function combines two terms: $\mathcal{L}_{\text{VAE}} =$

$\mathbb{E}_{q_\phi(\mathbf{z}|\mathbf{x})}[\log p_\theta(\mathbf{x}|\mathbf{z})] - \beta \cdot \text{KL}(q_\phi(\mathbf{z}|\mathbf{x})||p(\mathbf{z}))$ where the reconstruction term (MSE loss) ensures little information is lost, while the KL divergence term regularizes the latent space to be smooth and well-structured. We employ a sigmoid-based schedule for $\beta$, starting small with $\beta = 0.5$ to prioritize reconstruction accuracy, and ramping up to $\beta = 1.25$ to encourage disentangled representations. This scheduling strategy reduces the common posterior collapse problem in VAE training while ensuring high-quality reconstructions. Empirically this outperforms training with fixed $\beta$ values (see Table 5). The encoder-decoder is pretrained on $\sim$ 50 million sequences from the GIFT-Eval Pretrain dataset (Aksu et al., 2024), which comprises of $\sim$ 230 billion time series points across 88 diverse domains, including healthcare, finance, and industrial sensors. This large-scale pretraining ensures our representations generalize across domains, temporal resolutions, noise levels, and time horizons.

Overall, this design successfully balances three competing objectives: (1) capturing sufficient local temporal patterns within each patch, (2) maintaining computational efficiency through reasonable sequence lengths, and (3) preserving fine-grained information necessary for precise analysis.

### 3.2. TsLLM Architecture

Having established how we encode time series into dense representations, we now describe how these representations are integrated with a pretrained LLM to create a unified multimodal model. Our architectural design faces a fundamental challenge: how to combine time series and natural language modeling without degrading either modality's performance.

**LLM Integration** Rather than training a multimodal model from scratch, which is computationally infeasible, we build upon existing pretrained LLMs. We designed TsLLM to be model-agnostic, compatible with any decoder-only transformer architecture. In our experiments (Sect. 5), we demonstrate strong results with Qwen2.5 (Qwen et al., 2024), Llama 3.2 (Grattafiori et al., 2024), and SmolLM2 (Allal et al., 2025), showing that our approach generalizes across different model families and scales.

**Adapter Layers** The adapter layers serve as the critical bridge between our time series representations and the LLM's embedding space. These components must solve a challenging alignment problem: mapping from the encoder's learned representations (optimized for reconstruction) to the LLM's token embeddings (optimized for next-token prediction). Our encoder adapter architecture utilizes cross-attention between the patch encodings and a set of learned embeddings. Formally, we compute $y = \text{softmax}((xq)k^T)v$ where $x \in \mathbb{R}^{n \times d_t}$ is the input sequence of $n$ tokens, $q \in \mathbb{R}^{d_t \times d_L}$ is a linear mapping to the embedding space of the LLM, and $k, v \in \mathbb{R}^{z \times d_L}$ are the $z$ learnable key-value embedding pairs. Essentially, this

maps each patch embedding of the input time series to a (dynamic) convex combination of learned value embeddings. The key and value embeddings are initialized to a sample of $z = 1024$ embeddings from the pretrained LLM's input embedding layer. If frozen, this would mean the time series tokens are mapped directly into the space of the LLM's text embeddings, minimizing the domain gap, and the training just allows it to learn this mapping between domains better. The decoder adapter layer uses a more traditional MLP architecture.

**Scale-Shape Decomposition**  Time series in real-world applications span vastly different scales, e.g., heartbeats measured in millivolts, stock prices in thousands of dollars, and particle counts in millions. This variability presents unique challenges for multimodal models. Unlike traditional time series models that can normalize inputs and rescale outputs using known statistics, TsLLM must: (1) understand scale-dependent semantics (e.g., a heart rate of 200 bpm indicates tachycardia), and (2) generate time series at appropriate scales from text-only prompts (e.g., predict stock prices given a news report). Our solution decouples each time series into shape (normalized patterns) and scale (magnitude information) components. While the encoder processes normalized sequences $\hat{\mathbf{x}} = (\mathbf{x} - \mu_{\mathbf{x}})/\sigma_{\mathbf{x}}$ to extract shape features, we separately encode the normalization parameters $\mu_{\mathbf{x}}$ and $\sigma_{\mathbf{x}}$ to preserve scale information. We apply a log transformation to compress the scale range while preserving relative differences. The transformed parameters are encoded through dedicated MLPs and combined with the shape representations via addition. During generation, the model predicts both shape and scale for each token. We apply inverse transformations to recover the original scale, enabling generation of appropriately scaled time series.

**Model Fusion**  Integrating time series representations into the LLM requires careful handling of the token vocabulary and embedding spaces. We extend the LLM's vocabulary with a special "`<|ts|>`" token that acts as a placeholder for time series patches. During preprocessing, we identify time series segments in the input and replace each patch with a "`<|ts|>`" token, then in the embedding step we replace "`<|ts|>`" token embeddings with their corresponding time series representations. The embedding step is then given by:

$$\mathbf{e}_i = \begin{cases} \text{embed}(t_i) & \text{if } t_i \neq \texttt{<|ts|>} \\ \mathbf{h}_{\text{adapted},j} + \mathbf{s}_j & \text{if } t_i = \texttt{<|ts|>} \end{cases}$$

where $\mathbf{h}_{\text{adapted},j}$ is the $j$-th adapted time series representation and $\mathbf{s}_j$ is its corresponding scale embedding. For generation, we implement the same process in reverse: TsLLM is trained to predict "`<|ts|>`" tokens where time series is present, and we extract the corresponding hidden states for decoding. This design enables truly interleaved generation, meaning the model can freely mix text and time series in its outputs based on the task requirements.

## 3.3. Training TsLLM

In this section, we describe how TsLLM is trained to handle the full spectrum of time series analysis tasks. Our training strategy must address several challenges: unifying diverse task formats, balancing modality-specific losses, and ensuring stable multi-stage optimization.

**Unified Task Formulation**  A key feature of TsLLM is reformulating all time series tasks as autoregressive token generation problems over sequences of mixed text and time series tokens. This unification enables knowledge transfer across tasks and eliminates the need for task-specific or dataset-specific heads, simplifying training and inference. Figure 2 provides example prompts and responses for several tasks to demonstrate how this is done. Question-answering, report generation, and other text generation tasks require no special handling. Forecasting is done through autoregressive time series patch generation; during inference, each decoded patch is appended to the input and fed back into the model to predict the next patch. Classification maps onto constrained text generation; the model generates class labels directly as text. We use constrained decoding during inference to ensure outputs match valid class names. Anomaly detection can employ two approaches: direct detection, where the model outputs anomaly locations as text (e.g. "anomalies detected at timestamps 145-162, 203-210"), and reconstruction-based detection, where the model generates a time series of the expected normal behavior, with deviations indicating anomalies. Other tasks, including segmentation, point-level classification, and imputation, can similarly be formulated as time series generation tasks.

**Multi-Objective Training**  Our training objective balances three components: $\mathcal{L}_{\text{total}} = \lambda_1 \mathcal{L}_{\text{CE}} + \lambda_2 \mathcal{L}_{\text{MSE}} + \lambda_3 \mathcal{L}_{\text{scale}}$. The cross-entropy loss $\mathcal{L}_{\text{CE}}$ applies to text tokens, maintaining language modeling capabilities. The MSE loss $\mathcal{L}_{\text{MSE}}$ supervises time series reconstruction through the decoder. The scale loss $\mathcal{L}_{\text{scale}}$ ensures accurate recovery of normalization parameters.

**Staged Training Strategy**  The training process for TsLLM employs a carefully orchestrated three-stage approach, with each stage serving a specific purpose in building the model's capabilities. The staged approach proved crucial for stability as early experiments with all parameters unfrozen from the start tended to have loss spikes and performance plateaued early. *Stage 1: Alignment (1B tokens)* – We begin by training only the adapter and scale encoding layers while keeping both the encoder-decoder and LLM frozen. This stage uses raw time series sequences and simple synthetic tasks (estimating the mean, translating to and from text-based time series encodings, etc.) to establish the connection between modalities. By limiting the trainable parameters, we ensure stable alignment without

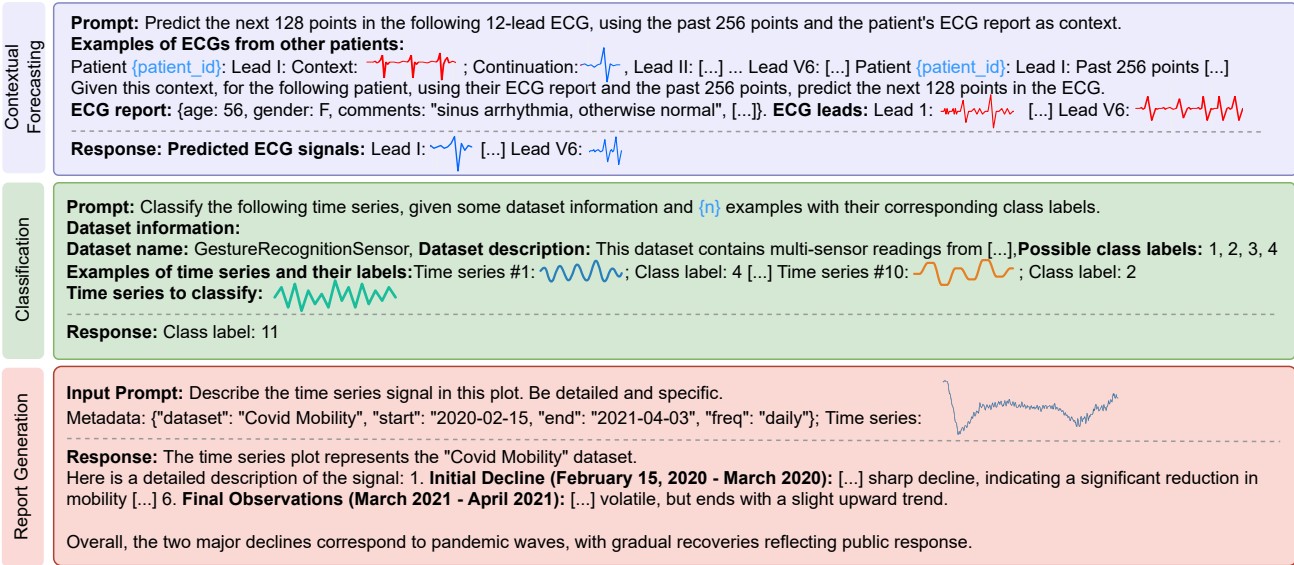

*Figure 2.* Illustrative example prompt-response pairs from the training dataset that demonstrate how TsLLM can interleave text and time series segments into a single sequence. The time series segments are plotted in this figure for clarity, but are represented as a sequence of embeddings in the model. Examples demonstrate versatility across tasks like: contextual forecasting (top), classification (middle), and report generation (bottom).

disrupting pretrained representations. *Stage 2: Continued Pretraining (9B tokens)* – With alignment established, we unfreeze all parameters for comprehensive multimodal training. This stage samples from the full training dataset, with conversational prompt-response data undersampled and reformatted as unstructured sequences. *Stage 3: Supervised Fine-Tuning (20B tokens)* – The final stage shifts to a conversational format, training on instruction-response pairs with loss computed only on responses. This stage transforms TsLLM from a base model for next-token prediction into an instruction-following model that can solve time series analysis tasks specified in natural language.

### 3.4. Data Curation and Synthesis

Training TsLLM to bridge time series and language modalities requires comprehensive data coverage across three categories: pure time series for encoder pretraining, paired time series with text for multimodal alignment, and synthetic tasks incorporating time series with natural language instructions and context for capability development. Our training corpus comprises approximately 27 billion tokens across 94 million examples, carefully curated to address the scarcity of high-quality time series-text pairs while ensuring broad domain coverage. Extensive work went into collecting and synthesizing sufficient high-quality data for training. This broadly included collecting and processing existing datasets (many with only time series, or time series plus labels), augmenting these datasets by connecting the signals with relevant text, and converting them into prompt-response pairs. It also involved generating synthetic prompt-response

pairs for simple tasks and general QA, which make up a large portion of the training tokens. An in-depth description of the complete data gathering process is provided in Appendix B.

## 4. Implementation Details

The patch encoder and decoder each employed a 16-layer MLP with SwiGLU, LayerNorm, embedding size 128, and patch size 32. The VAE uses a 12-dimensional latent space with KL divergence scaled by $\beta$ starting at 0.5 and ramping up to 1.25, trained for 50M samples (batch size 2048) using AdamW (peak LR 5e-4) and MSE reconstruction loss. TsLLM was optimized using AdamW (peak LR 5e-5) with cosine scheduling, linear warmup, and gradient clipping at 1.0. Training used 30B tokens sampled from a 27B token dataset with max sequence length 4096, applying MSE loss for time series prediction and cross-entropy for language tokens. We use Qwen2.5-7B (Qwen et al., 2024) as the base LLM. Training required approximately 800 GPU-hours on four NVIDIA H100s. The added components total 6.7M parameters (1M each for encoder/decoder with scale modules, 3.7M for encoder adapter, 1M for decoder adapter), which is small relative to the base LLM.

## 5. Results

To demonstrate TsLLM's capabilities at the intersection of time series analysis and natural language understanding, we evaluate it primarily on time series tasks that require

language integration – question-answering and contextual forecasting. We provide additional results on traditional time series tasks (classification, forecasting, anomaly detection) in Appendix A to establish that it can achieve competitive performance with specialized models on these types of tasks even if it does not outright beat them. This evaluation strategy reflects our aims: while we show competitive performance on traditional tasks, our primary focus is demonstrating capabilities that standard approaches cannot provide.

## 5.1. Evaluations

**Question-Answering** To evaluate the time series reasoning and question-answering capabilities of TsLLM, we assess its performance using the electrocardiogram question-answering (ECG-QA) benchmark (Oh et al., 2023) comprised of ECG recordings, questions, and answers across several question types: yes/no verification (S-verify), choose between options (S-choose), and open-ended queries (S-query). We compared TsLLM performance based on exact match accuracy to top-performing specialized models trained on the ECG-QA dataset, namely $M^3AE$ (Chen et al., 2022), Q-Heart (Pham et al., 2025), and ECG-LM (Yang et al., 2025a). Each of these models is explicitly designed and trained for ECG-QA, whereas TsLLM is exposed to ECG-QA during training, but only a small fraction (<4%) of its training tokens originated from this dataset, and its design is not specialized for ECG-QA. We also compare it to several LLMs and two time series LLMs, none of which were trained on ECG-QA, but receive few-shot examples in context. As Table 1 illustrates, despite the lack of fine-tuning, the models perform well in the category, and TsLLM (few-shot) earns the best or the second-best performance across all queries. While the table focuses on a subset of the question types from this dataset, TsLLM can be applied to other question types with the proper prompt format, demonstrating the ability of the models to handle complex tasks that require expertise.

*Table 1.* Question-answering exact-match accuracy comparison on ECG-QA by question type. Models with the "❄" icon have not been trained at all on ECG-QA, but received few-shot examples as context. Models with the "🔥" icon were trained on the training set of ECG-QA. We trained variants of TsLLM with and without training on ECG-QA for fair comparisons.

| Method | Question Type | | | |
|---|---|---|---|---|
| | *S-Verify* | *S-Choose* | *S-Query* | **Average ↑** |
| Gemini 2.5 Pro ❄ | 55.7 | 30.4 | 17.6 | 30.9 |
| GPT o4-mini ❄ | 50.0 | 47.8 | 26.5 | 45.9 |
| ChatTime ❄ | 33.3 | 2.5 | 5.0 | 11.5 |
| ChatTS ❄ | 56.7 | 32.5 | 15.0 | 29.1 |
| **TsLLM** (no ECG-QA) ❄ | **68.5** | **50.6** | **32.7** | **50.8** |
| $M^3AE$ 🔥 | 74.6 | 57.1 | 41.0 | 57.6 |
| Q-Heart 🔥 | **90.9** | 60.3 | 32.9 | 61.4 |
| ECG-LM 🔥 | 75.8 | 57.4 | 39.9 | 57.7 |
| **TsLLM** 🔥 | 85.3 | **60.8** | **49.0** | **65.0** |

We also evaluate TsLLM on TimeSeriesExam (Cai et al., 2024), a multiple-choice question answering benchmark that tests LLMs' time series understanding across diverse settings. TsLLM performs far better than even frontier models many times larger than it, demonstrating the effectiveness of our encoding and training approach.

*Table 2.* Accuracy on TimeSeriesExam, a synthetic time series question-answering benchmark.

| Model | Accuracy ↑ |
|---|---|
| Qwen 2.5 7B ❄ | 29.8% |
| Gemini 2.5 Pro ❄ | 29.5% |
| GPT o4-mini ❄ | 49.1% |
| ChatTime ❄ | 41.9% |
| ChatTS ❄ | 56.4% |
| **TsLLM** ❄ | **86.4%** |

**Contextual Forecasting** We evaluate the ability of TsLLM to forecast time series using the Context Is Key (CiK) dataset (Williams et al., 2025), which contains 71 realistic forecasting tasks spanning 7 domains specifically designed to require understanding and integrating textual information for successful prediction. We do not use the standard evaluation metric (continuous ranked probability score) of the CiK benchmark as TsLLM does not directly support uncertain probabilistic forecasting, and we train on a held-out subset of problems sampled from CiK, so results are not directly comparable with the original benchmark, but it still provides a fair and valuable assessment of TsLLM's ability to perform contextual forecasting. TsLLM outperforms both traditional ML methods (which cannot use the context) as well as frontier LLMs (which can use the context, but have poor time series understanding) by a significant margin, demonstrating that it is able to bridge the gap between numerical signals and text-based context far better than existing methods.

*Table 3.* Contextual forecasting performance on the Context Is Key benchmark.

| Model | SMAPE (%) ↓ | MASE ↓ |
|---|---|---|
| Linear Regression 🔥 | 75.4 | 101.5 |
| XGBoost 🔥 | 76.8 | 80.2 |
| ARIMA 🔥 | 90.7 | 134.4 |
| Qwen 2.5 7B ❄ | 92.6 | 139.2 |
| Gemini 2.5 Pro ❄ | 90.8 | 98.1 |
| GPT o4-mini ❄ | 72.6 | 70.5 |
| ChatTime ❄ | 112.6 | 92.3 |
| ChatTS ❄ | 119.4 | 204.0 |
| **TsLLM** 🔥 | **64.5** | **64.7** |

## 5.2. Ablation Studies

We conduct ablation studies on our methodology at a smaller scale, using SmolLM2-360M as the pretrained LLM backbone, with results in Table 4. Performance is measured by

*Table 4.* Ablation study over the components of TsLLM, as measured by downstream average accuracy on ECG-QA.

| Ablation Category | Value | Accuracy (%) ↑ | Change (%) |
|---|---|---|---|
| Baseline | | 59.6 | – |
| LLM | SmolLM2 135M | 58.4 | -2.0% |
| LLM | SmolLM2 1.7B | 60.8 | +2.0% |
| LLM | Llama 3.2 3B | 62.5 | +4.9% |
| LLM | Qwen 2.5 7B | 65.0 | +9.1% |
| Pretraining | None (random init) | 43.7 | -26.7% |
| Adapter | MLP | 59.3 | -0.5% |
| Adapter | Linear | 55.8 | -6.4% |
| Encoder | VQ-VAE | 52.4 | -12.1% |

accuracy on ECG-QA. These results demonstrate that the size and performance of the backbone LLM matter significantly, suggesting that TsLLM can be scaled up further to improve performance. They also show that simpler adapter layers perform somewhat worse than our cross attention based approach. Finally, using an encoder-decoder pair with quantized latent embeddings currently has a substantial negative impact on performance, highlighting the importance of precise, high-fidelity encoder and decoder models.

*Table 5.* Trade-off between reconstruction quality and downstream task performance for different encoder-decoder configurations. Color coding: green indicates improvement, red indicates degradation relative to baseline.

| Ablation | Baseline | Patch size | | $\beta$ | | Objective | | # of layers | |
|---|---|---|---|---|---|---|---|---|---|
| Value | | 16 | 64 | 1.0 | 1.8 | AE | VQ-VAE | 12 | 48 |
| Recon. MSE ↓ | 0.217 | 0.151 | 0.308 | 0.173 | 0.367 | 0.005 | 0.432 | 0.217 | 0.216 |
| % Change | – | -30% | +42% | -20% | +69% | -97% | +99% | 0% | -1% |
| ECG-QA Acc. ↑ | 55.9 | 54.1 | 53.5 | 55.5 | 55.3 | 54.0 | 51.8 | 55.6 | 54.9 |
| % Change | – | -3.2% | -4.3% | -0.7% | -1.1% | -3.4% | -7.3% | -0.5% | -1.8% |

**Time Series Encoder-Decoder Training** An ablation study on time series encoder-decoder reconstruction performance (Table 5) reveals a critical trade-off between the reconstruction performance of the encoder-decoder pair and downstream performance on practical tasks. We look at the MSE on our VAE validation data across encoder-decoder pairs with different hyperparameter settings, as well as the accuracy of small-scale implementations of TsLLM (SmolLM2-135M, 10B training tokens) trained using each of these pairs. This table makes clear that it is possible to improve the reconstruction ability of the VAE significantly by reducing the patch size or $\beta$, but this leads to more granular representations of the time series that are not as informative or effective when integrated into TsLLM. The representation abstraction can also be increased by increasing $\beta$ or selecting a larger patch size, but this hurts reconstruction, which also hurts performance of TsLLM. We have optimized for a balance between these objectives in order to maximize performance of TsLLM on real-world tasks.

### 5.3. Performance on Text Generation Tasks

While we have extended the capabilities of the LLM to include time series analysis, our aim is to build a general-purpose model with the same level of intelligence and ca-

pabilities as the original pretrained LLM – in this case, Qwen2.5 7B. To this end, we evaluate the performance of TsLLM on standard text generation benchmarks against a fine-tune of Qwen2.5 7B that trains the model on the same text-only datasets we used for training TsLLM. We find nearly identical performance on all benchmarks we tested – specifically MMLU-Pro (53.5 vs 53.5), LiveBench (32.4 vs 32.9), GPQA (36.3 vs 36.4), MATH (68.9 vs 68.1), and GSM8K (89.1 vs 89.2). This indicates that adding time series analysis capabilities to the LLM does not significantly degrade its performance on standard text generation tasks.

## 6. Discussion

TsLLM augments pretrained LLMs with time series perception, enabling analysis that combines temporal reasoning with natural language understanding. This approach parallels vision-language models, which proved valuable by enabling new capabilities rather than by surpassing specialized computer vision models on traditional benchmarks. Similarly, TsLLM's contribution lies in enabling capabilities that traditional approaches cannot provide: answering questions about temporal patterns, incorporating unstructured context into predictions, and generating natural language explanations.

The contextual forecasting and QA results (Tables 1,2,3) demonstrate that neither frontier LLMs nor traditional time series models can bridge this gap alone. LLMs fail to perceive temporal signals accurately while specialized models cannot leverage textual context, explaining TsLLM's substantial margins over both. Our ablations further show that performance scales with LLM backbone size, suggesting future base model improvements will directly benefit TsLLM without architectural changes. Notably, optimizing reconstruction fidelity hurts downstream performance; the $\beta$-VAE's regularized representations outperform a near-perfect autoencoder, indicating the LLM benefits from semantic abstraction over signal-level precision.

Our approach is complementary to specialized time series models rather than competitive with them. TsLLM does not aim to replace established methods for pure numerical prediction tasks, where specialized models remain effective and efficient. Instead, it addresses problems requiring integration of temporal analysis with contextual understanding, domain knowledge, and natural language interaction.

**Limitations** Several limitations of TsLLM warrant discussion. First, TsLLM requires substantially more computational resources than traditional time series models, both for training and inference. This is an expected tradeoff when leveraging the capabilities of LLMs. The benefit is that TsLLM can be trained once and applied to diverse tasks through in-context learning, amortizing costs across applications and eliminating the need for task-specific model

development. Second, while TsLLM demonstrates competitive performance on traditional benchmarks, it does not achieve state-of-the-art results on pure numerical tasks like forecasting or anomaly detection. Our goal is not to compete with specialized models optimized for specific metrics, but to enable capabilities that those models cannot provide. We argue that the performance gap on traditional benchmarks is acceptable given TsLLM's unique ability to incorporate contextual information, go beyond the standard time series tasks, and answer questions. Finally, evaluation methodology for time series-language tasks lacks standardization. Unlike pure time series or pure language tasks, there are no established benchmarks for assessing models that bridge both modalities. Developing rigorous, expert-validated benchmarks for tasks like time series question-answering and contextual forecasting should be a priority for the field.

## 7. Conclusion

This work introduced TsLLM, a time-series-augmented large language model that enables unified reasoning over interleaved temporal signals and natural language. By combining a high-fidelity patch-based encoder-decoder, scale-aware representations, and multimodal alignment with a pretrained LLM, TsLLM addresses a key limitation of existing language models: their inability to efficiently and accurately perceive dense numerical time series. Our results show that this added temporal perception unlocks capabilities beyond conventional time series models and text-only LLMs, including time series question-answering, contextual forecasting, natural language explanation, and flexible few-shot adaptation across domains.

TsLLM is not intended to replace specialized models for narrowly defined numerical tasks. Instead, it expands the scope of time series analysis to settings where prediction, interpretation, context, and communication must be handled together. The strong performance gains on language-integrated time series benchmarks, together with minimal degradation on standard text tasks, suggest that augmenting LLMs with dedicated temporal interfaces is a promising path toward more general, interactive, and context-aware systems for real-world time series understanding.

## Impact Statement

This paper presents work whose goal is to advance the field of machine learning. There are many potential societal consequences of our work, none of which we feel must be specifically highlighted here. See Appendix D for a more in-depth analysis of the broader impacts of this work.

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

# A. Additional Results

In this section we present additional empirical results as a continuation from Section 5. Figure 3 plots the relationship between the patch size set for the time series encoder and decoder, and the resulting reconstruction error (MSE) of the VAE model on a subset of the LOTSA dataset. It demonstrates that for a fixed representation size, patch size and reconstruction performance are inversely correlated, so smaller patch sizes have inherent benefits. However, they also increase the sequence length of a time series, shifting complexity to the LLM, which can harm downstream performance. Finding the right balance between these factors is critical for optimizing capabilities across a range of tasks and datasets.

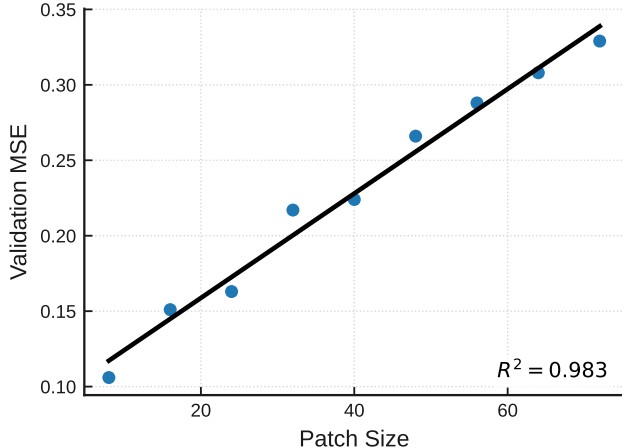

*Figure 3.* The relationship between patch size for the time series encoder and reconstruction MSE for the VAE on the validation set.

## A.1. Classification

For classification tasks, we evaluate our model on the UCR-TSC dataset. Performance is assessed using the mean and median accuracy across sub-datasets, ensuring robustness. Results can be found in Table 6. TsLLM achieves competitive performance (81.2% mean accuracy with few-shot prompting) without task-specific architectures, demonstrating effective transfer of language model capabilities to temporal pattern recognition.

*Table 6.* Time series classification accuracy on the UCR-TSC benchmark collection. Although TsLLM is not designed for general numerical-only tasks, it performs comparably with top models.

| Metric | DTW | CNN | T-Loss | TS2Vec | Times-Net | MOM-ENT0 | TsLLM (1S) | TsLLM (FS) |
|---|---|---|---|---|---|---|---|---|
| Mean | 0.764 | 0.751 | 0.833 | **0.851** | 0.572 | 0.794 | 0.792 | 0.812 |
| Median | 0.768 | 0.773 | 0.849 | **0.871** | 0.565 | 0.815 | 0.798 | 0.823 |

## A.2. Forecasting

We evaluate forecasting performance on the GIFT-Eval dataset. Performance is measured using a relative Mean Absolute Percent Error (rMAPE) normalized against the seasonal naive baseline. All baseline results are as reported in Aksu et al. (2024). We further break down performance by the number of variates (univariate vs multivariate) and prediction length, as illustrated in Table 7, where TsLLM outperforms competitively with other models although not completely outperforming all.

**Anomaly Detection**   We also evaluate TsLLM on anomaly detection, specifically on the MSL dataset (Hundman et al., 2018), as shown in Table 8. While our approach achieves a competitive F1 score of 73.3, there remains a performance gap compared to state-of-the-art models like UniTS-FT (81.2) and iTransformer-FT (80.4). The results demonstrate TsLLM's versatility across different time series tasks, though specialized models may still hold advantages in specific domains.

*Table 7.* Forecasting performance on GIFT-Eval benchmark across different data characteristics. Values show relative MAPE (lower is better) normalized to seasonal naive baseline, following (Aksu et al., 2024). TsLLM with few-shot prompting (FS) achieves best-in-class performance on univariate series (0.78) while remaining competitive overall (0.87), with particularly strong results on long-horizon forecasts.

| Subset | Naive | Seasonal Naive | TFT | PatchTST | iTransformer | TimesFM | VisionTS | Chronos Base | Chronos Large | Moirai Base | Moirai Large | TsLLM | TsLLM (F-S) |
|---|---|---|---|---|---|---|---|---|---|---|---|---|---|
| GIFT-Eval | | | | | | | | | | | | | |
| All | 1.20 | 1.00 | 1.12 | **0.86** | 0.98 | 1.25 | 0.98 | 0.94 | 0.93 | 1.01 | **0.86** | 0.89 | 0.87 |
| Univariate | 1.19 | 1.00 | 0.89 | 0.85 | 0.85 | 0.87 | 0.93 | 0.81 | 0.80 | 0.95 | 0.81 | 0.82 | **0.78** |
| Multivariate | 1.21 | 1.00 | 1.40 | **0.88** | 1.15 | 1.71 | 1.04 | 1.11 | 1.10 | 1.09 | 0.93 | 0.97 | 0.99 |
| L Horizon | 1.35 | 1.00 | 1.16 | **0.96** | 0.99 | 1.80 | 1.03 | 0.99 | 1.00 | 1.09 | 1.01 | 1.00 | **0.96** |
| M Horizon | 1.30 | 1.00 | 1.26 | **0.90** | 1.02 | 1.52 | 1.02 | 1.38 | 1.33 | 1.11 | 0.96 | 0.96 | 0.93 |
| S Horizon | 1.10 | 1.00 | 1.05 | 0.80 | 0.97 | 0.93 | 0.95 | **0.75** | **0.75** | 0.94 | 0.77 | 0.83 | 0.82 |
| ETTh1 48 | 1.62 | 1.00 | 0.94 | 0.78 | 0.81 | 0.84 | 0.93 | **0.76** | 0.83 | 0.95 | 0.87 | 0.90 | 0.83 |
| M4 hourly 48 | 2.42 | 1.00 | 1.87 | 1.07 | 1.13 | 0.65 | 0.81 | **0.54** | **0.54** | 0.82 | 0.68 | 0.74 | 0.65 |
| Electricity 480 | 1.88 | 1.00 | 0.89 | 0.75 | 0.84 | 1.15 | 0.79 | 0.82 | 0.82 | 0.94 | 0.79 | 0.79 | **0.73** |
| Electricity 720 | 2.58 | 1.00 | 0.77 | 0.75 | 0.91 | 1.25 | 0.92 | 0.87 | 0.87 | 0.95 | 0.86 | 0.78 | **0.74** |

*Table 8.* Anomaly detection performance on MSL.

| Model | MSL (F1 score) |
|---|---|
| Anomaly Trans. | 78.0 |
| TimesNet-*FT* | 33.9 |
| iTransformer-*FT* | 80.4 |
| PatchTST-*FT* | 79.9 |
| UniTS-*PMT* | 75.4 |
| UniTS-*FT* | 81.2 |
| TsLLM | 73.3 |

# B. Datasets

Our training approach leverages a large collection of both real-world datasets and synthetic data, spanning many domains and time series analysis tasks. We strategically augment existing datasets with synthetic data generation to ensure broad coverage of time series understanding and prediction capabilities while maintaining data quality and diversity. This section provides detailed descriptions of our data sources, organized by modality and generation approach.

### B.1. Training Data Overview

Table 9 summarizes the key statistics for all datasets used in training TsLLM. Our training corpus encompasses nearly 27 billion total tokens across 20 distinct datasets, spanning diverse domains including medical signals, financial data, environmental measurements, and industrial sensors. The datasets support multiple task types including forecasting, classification, anomaly detection, segmentation, and question-answering, enabling TsLLM to develop comprehensive time series analysis capabilities.

### B.2. Unimodal Time Series Data

Our approach leverages a diverse collection of unimodal time series datasets spanning multiple domains and analytical tasks. These datasets provide the foundation for training TsLLM's time series understanding capabilities across both unlabeled pretraining data and task-specific labeled benchmarks.

**Large-Scale Pretraining Data**   For large-scale pretraining, we utilize the GIFT-Eval Pretraining dataset (Aksu et al., 2024), a collection of 4.5 million time series across 88 datasets spanning seven domains. GIFT-Eval provides a dedicated split for pretraining that is separate from its training and testing sets, making it particularly suitable for developing and evaluating general-purpose time series models. The benchmark's temporal granularity ranges from secondly to yearly frequencies, incorporating both univariate and multivariate sequences with forecast horizons varying from short-term to

| Name | Domain | Tasks | Total Tokens | Response Tokens | Text/TS Tokens | Samples |
|---|---|---|---|---|---|---|
| GIFT-Pretrain | Multi-Domain | Forecasting | 7,951,924,108 | 3,861,122,236 | (1,990,801,872/5,961,122,236) | 74,805,850 |
| ECG-QA | ECG | Q-A | 1,071,939,892 | 10,119,218 | (468,584,812/603,355,080) | 942,950 |
| News-Stocks | Finance | Forecasting | 124,179,199 | 3,464,920 | (118,277,891/2,436,388) | 61,922 |
| PTBXL-Classification | ECG | Classification | 20,615,930 | 296,710 | (3,577,588/16,741,632) | 21,799 |
| PTBXL-Forecasting | ECG | Forecasting | 22,482,365 | 5,740,733 | (0/16,741,632) | 21,799 |
| PTBXL-Report Gen | ECG | Report Gen. | 23,155,582 | 3,689,075 | (2,724,875/16,741,632) | 21,799 |
| Monster | Multi-Domain | Classification | 1,665,431,147 | 12,045,400 | (929,008,270/736,422,877) | 1,998,336 |
| TTC-Medical | Medical | Mixed | 628,367 | 232,387 | (390,936/5,044) | 330 |
| TTC-Climate | Climate | Mixed | 2,132,627 | 559,584 | (1,563,351/9,692) | 1,320 |
| UCR-TSC | Time Series | Classification | 68,966,571 | 797,925 | (50,634,942/17,533,704) | 60,555 |
| UTSD | Multi-Domain | Forecasting | 78,164,051 | 51,392,965 | (0/26,771,086) | 873,621 |
| Weather5K | Weather | Forecasting | 673,077,250 | 515,506,637 | (665,491,994/7,585,256) | 674,245 |
| UEA | Multi-Domain | Classification | 29,306,232 | 1,821,291 | (16,888,337/10,596,604) | 22,606 |
| Child Sleep Detct. | Sensor | Segmentation | 916,281 | 52,061 | (531,027/333,193) | 5,331 |
| PSM | Machine | Anomaly | 704,856 | 240,546 | (285,299/179,011) | 220,322 |
| MSL | Spacecraft | Anomaly | 893,631 | 304,969 | (361,708/226,954) | 132,046 |
| SMAP | Spacecraft | Anomaly | 1,731,267 | 590,829 | (700,751/439,687) | 562,800 |
| SWaT | Infrastructure | Anomaly | 6,106,917 | 2,084,106 | (2,471,848/1,550,963) | 944,920 |
| SMD | Machine | Anomaly | 6,624,762 | 2,260,831 | (2,681,452/1,682,479) | 1,416,825 |
| Synthetic | Multi-Domain | Mixed | 15,186,959,206 | 2,815,439,507 | (11,267,019,855/3,919,939,351) | 11,273,306 |
| **Total** | **All** | **All** | **26,935,940,241** | **7,287,761,930** | **(15,521,996,808/11,340,414,501)** | **94,062,682** |

*Table 9.* Comprehensive statistics for all datasets used in TsLLM training, showing the distribution of tokens across text and time series modalities, along with task diversity across domains.

long-term predictions. With over 230 billion data points, this dataset provides the scale and diversity necessary for training foundation models. We use it extensively for pretraining our time series encoder and decoder models, as well as training TsLLM for forecasting tasks, while carefully ensuring no data leakage into the GIFT-Eval test set used in our evaluation.

We also leverage two additional large-scale datasets for comprehensive coverage: the Unified Time Series Dataset (UTSD) from Tsinghua University's THUML lab, a large-scale collection spanning seven domains (Energy, Environment, Health, IoT, Nature, Transportation, and Web) with up to one billion time points that combines publicly available online data with empirical measurements from machine operations, and LOTSA from Salesforce, which encompasses approximately 27 billion observations across nine domains (Energy, Transport, Climate, CloudOps, Web, Sales, Nature, Economics/Finance, and Healthcare) with sampling frequencies ranging from seconds to years.

**Classification Benchmarks** For time series classification tasks, we employ two principal benchmark collections that have become standard in the field. The UCR Time Series Classification Archive (Dau et al., 2019) comprises 128 datasets spanning diverse applications including electrocardiograms, motion capture sequences, sensor readings, and spectrographs. These datasets vary significantly in length, class distribution, and domain-specific characteristics, providing comprehensive coverage for univariate time series classification. Complementing this, the UEA Time Series Classification Archive (Bagnall et al., 2018) extends our capabilities to multivariate time series, offering 30 datasets that capture complex temporal patterns in applications such as human activity recognition, motion capture, and handwriting analysis. Together, these repositories represent the gold standard for time series classification research, with the UCR archive focusing on univariate sequences and the UEA archive addressing more complex real-world scenarios where multiple variables are tracked concurrently.

**Specialized Task Datasets** For segmentation tasks, we employ the Child Mind Institute's "Detect Sleep States" dataset (Esper et al., 2023), which consists of approximately 500 multi-day accelerometer recordings captured from wrist-worn devices, with each series corresponding to a unique participant. The dataset provides expert annotations for sleep onset and wakeup events, enabling comprehensive evaluation of temporal segmentation capabilities in physiological monitoring applications.

For anomaly detection, we incorporate several benchmark datasets representing different operational contexts and domains. These include NASA's Mars Science Laboratory (MSL) and Soil Moisture Active Passive (SMAP) telemetry datasets (Hundman et al., 2018), which provide spacecraft sensor data with labeled anomalies; eBay's Pooled Server Metrics (PSM) dataset (Abdulaal et al., 2021), containing server performance metrics; the Server Machine Dataset (SMD) (Su et al., 2019), which includes machine operation data; and the Secure Water Treatment (SWaT) dataset (Mathur and Tippenhauer, 2016), representing critical infrastructure monitoring. This diverse collection ensures robust anomaly detection capabilities across industrial, space, and cybersecurity applications.

**Medical Time Series Data** We also utilize the PTB-XL dataset (Wagner et al., 2022; Goldberger et al., 2000), which consists of 21,799 clinical 12-lead ECGs collected from 18,869 patients. Each recording spans 10 seconds and includes

expert annotations from up to two cardiologists, using 71 distinct SCP-ECG standard statements. The dataset includes comprehensive metadata including patient demographics and diagnostic information, making it particularly valuable for developing medical time series analysis capabilities. From this dataset, we developed specialized variants optimized for different tasks: one for time series classification tasks and another for time series forecasting applications, as detailed in our data synthesis section.

**Text-Only Training Data**   To ensure TsLLM maintains strong text generation capabilities alongside its time series analysis skills, we incorporate two standard text datasets into our training mixture. We utilize a subset of FineWeb-Edu (Lozhkov et al., 2024), a large-scale educational text corpus containing high-quality educational content curated from various web sources and specifically filtered for instructional content, explanations, and academic materials. Additionally, we include SmolTalk (Allal et al., 2025), a synthetic dataset of 1M instruction-following examples created for supervised fine-tuning, covering tasks like summarization and reasoning. This combination ensures that our model's natural language capabilities remain robust while learning time series-specific skills.

## B.3. Paired Time Series and Text Data

Multimodal datasets that combine time series with textual information are essential for training TsLLM's ability to reason about temporal patterns in context. Our approach leverages both existing paired datasets and novel synthetic generation techniques to create comprehensive training data for time series-text understanding.

**Existing Multimodal Datasets**   Existing datasets that pair time series with textual data have been instrumental in developing models that can bridge numerical and linguistic understanding. The ECG-QA dataset (Oh et al., 2023) provides a specialized medical dataset that pairs electrocardiogram readings with natural language questions and answers about cardiac conditions and diagnoses. Built from PTB-XL (Wagner et al., 2022) and MIMIC-IV-ECG (Gow et al., 2023) databases, it includes both templated and naturally-phrased questions that test understanding of ECG features like heart rhythm patterns.

The TextTime Corpus (TTC) (Kim et al., 2024) offers broader domain coverage by pairing two types of temporal data: (1) daily weather measurements with forecast discussions and (2) patient vital signs paired with clinical notes from MIMIC-III (Johnson et al., 2016). This diversity in both domains and text types provides valuable training examples for contextual reasoning about time series data.

Time-MMD (Liu et al., 2024) extends multimodal capabilities by combining numerical and textual sequences across 9 domains, incorporating corresponding reports and search results that provide rich contextual information for time series analysis.

**New Dataset Construction**   For our data collection efforts, we focused particularly on the financial domain, creating a new dataset that combines S&P 500 daily closing prices with corresponding news articles. This pairing enables exploration of contextual forecasting tasks where models must consider both numerical trends and relevant textual information when making predictions.

Our financial dataset construction process involves several key steps: first, we collect historical S&P 500 price data; second, we gather contemporaneous news articles that might impact market movements; and finally, we align these data sources temporally to create paired examples suitable for contextual forecasting tasks.

We constructed this dataset using news data from two sources with different granularities: The Bloomberg dataset (Genloop, 2024) provides full articles with multiple stock symbol associations, while the Benzinga dataset (Aenlle, 2020) contributes headlines with single stock symbol associations. The dataset includes daily closing prices and company fundamentals (sector, industry, country, and business descriptions). Each forecasting task combines historical price data and news context (up to 4 Bloomberg articles or 16 Benzinga headlines) to predict a 32-day forward price sequence. We employ a minimum context window of 32 days with prediction windows spaced at 16-day intervals, ensuring all contextual information precedes the target sequence.

## B.4. Synthetic Data Generation

To ensure comprehensive coverage of time series analysis tasks and provide sufficient training data for multimodal understanding, we developed a systematic data synthesis approach encompassing three main components: augmentation of existing time series datasets, foundational text generation tasks, and large-scale question-answering synthesis.

**PTB-XL Dataset Augmentation** We created three synthetic variants of the PTB-XL dataset to expand its applicability beyond traditional classification tasks. Using Qwen3-VL-235B (Bai et al., 2025), we systematically generated textual content paired with the existing ECG recordings.

The first variant, designed for contextual forecasting, augments each ECG with multiple generated reports using different templates including basic summaries, highlight-focused reports, and interpretative reports with clinical context. Each template was designed to handle both English and German inputs while maintaining English output consistency. The second variant, optimized for classification tasks, pairs each ECG with demographic information (age, sex, height, weight) while excluding statement fields and signal-derived features. This cleaned subset ensures models must rely primarily on raw signal patterns rather than pre-computed features, providing a more challenging and realistic evaluation scenario.

The third variant supports report generation tasks by combining ECG signals with reports generated under specific constraints. These include structured responses containing only signal-derived information, narrative reports presenting findings as flowing clinical stories, and interpretative reports providing broader clinical context. Each record includes both the generation prompt and the resulting report, enabling comprehensive evaluation of medical time series understanding and generation capabilities.

**Foundational Time Series Tasks** To establish and evaluate basic time series manipulation capabilities, we created a synthetic instruction-following dataset using time series sampled from the GIFT-Eval pretraining split. Employing various state-of-the-art LLMs, we generated diverse instruction-response pairs across four fundamental tasks that serve as building blocks for more complex time series operations.

The first task involves encoding time series as text, where numerical sequences are converted to comma-separated string representations with consistent precision and formatting. The complementary second task requires reconstructing time series from their textual encodings, testing the model's ability to accurately parse and interpret numerical sequences. The third task focuses on basic statistical computations, specifically calculating fundamental measures like mean and standard deviation of given time series. The fourth task extends this to comprehensive statistical analysis, requiring the model to compute and return a structured JSON object containing seven statistical measures including median, quartiles, and extrema.

This collection of foundational tasks serves as a warm-up phase in training, establishing essential capabilities in numerical manipulation, statistical analysis, and format conversion that underpin more sophisticated time series reasoning and analysis operations.

**Large-Scale Question-Answering Synthesis** To create a comprehensive training dataset for time series question-answering capabilities, we synthesized a large-scale dataset comprising over 2 million question-answer pairs, totaling approximately 10 billion tokens. This dataset was generated through a systematic combination of carefully curated question templates, diverse time series data, and responses from state-of-the-art language models. All time series data was sampled from the GIFT-Eval pretrain split, with rigorous filtering applied to ensure data quality and maintain consistent sequence lengths suitable for training.

For time series representation, we implemented two complementary encoding schemes to maximize model flexibility. The text encoding represents time series values as comma-separated numerical sequences with configurable precision and normalization options, accompanied by rich metadata including sampling frequency, domain information, and basic statistical summaries. The image encoding visualizes time series as line plots encoded as RGB images, enabling compatibility with vision-language models and providing an alternative modality for time series understanding.

Our question-generation system employs template-based sampling with carefully weighted instruction types and styles. The templates comprehensively cover various analytical tasks ranging from trend identification and pattern recognition to statistical analysis and anomaly detection. Each generated question is paired with relevant time series metadata and optional statistical summaries to provide sufficient context for accurate and informative responses. This approach ensures broad coverage of different analytical scenarios while maintaining consistency in question quality and complexity.

To generate high-quality responses, we employed a diverse ensemble of state-of-the-art language models including frontier text-only models (Qwen3-235B (Yang et al., 2025b), GLM-4.5 (GLM-4.5 Team et al., 2025)) and vision-language models (Qwen3-VL-235B (Bai et al., 2025), GLM-4.5V (GLM-V Team et al., 2025), Gemini 2.0 (Gemini Team et al., 2023)). While synthetic responses may contain some inaccuracies, recent findings (Bansal et al., 2025) suggest that larger quantities of moderate-quality training data can be more effective than smaller sets of perfect examples for LLM training. The resulting

scale and diversity of our dataset provides comprehensive coverage of different analytical scenarios, making it particularly suitable for training models in time series analysis and reasoning tasks.

## C. Data Samples

The training data consists of textual instructions followed by a time series sequence, marked with a special token. The corresponding labels provide the indices of sleep onset and wake-up points. Example training instances is shown below:

### Child Sleep Detection

INPUT

```
"Instruction: Identify sleep onset and wake-up points in the following time series.
Time series signal: <|ts|>"
```

LABEL

```
"onset indices": [252, 469]
"wakeup indices": [347, 610]
```

### GIFT-Eval

INPUT

```
<|begin_of_text|> Time series metadata: dataset = Monash M3 Quarterly,
start = 1977-01-01T00:00:00, end = 1993-07-01T00:00:00,
sample frequency = Q-DEC, length = 66 points
Time series signal: <|ts|>
<|end_of_text|>
```

### Synthetic QA

INPUT

```
<|begin_of_text|> You are a helpful AI assistant trained to understand numerical time series data.
Prompt: Describe how this time series might continue if extended.
Series metadata:
domain: Nature
dataset: Star Light Curves
Time series: <|ts|>
```

RESPONSE

```
The time series you've provided appears to represent the brightness (or magnitude) of a star over time,
    specifically from the dataset of star light curves. This kind of data is crucial in astrophysics for
    understanding the behavior of stars, such as their variability and periodicity.

Analysis of the Time Series:
1. Trend and Pattern: The series starts with a gradual decrease in brightness, reaching a minimum value, and then
    gradually increasing back to a higher brightness. This suggests a periodic or cyclic pattern, which is common
     in variable stars.
2. Periodicity: The pattern of decrease and increase repeats, indicating that the star might be a pulsating
    variable star (like a Cepheid variable or RR Lyrae star) or a star with an eclipsing companion (like an
    eclipsing binary system).
3. Amplitude: The amplitude of the variation is significant, ranging from approximately -0.57 to 2.80 in the
    magnitude scale. This is typical for stars with strong variability.
4. Duration of Cycle: The complete cycle from peak to trough and back to peak appears to be quite long, suggesting
     a long-period variable star.

Prediction of Future Behavior:
Given the periodic nature of the time series, we can predict that the pattern will likely continue in a similar
    fashion:
1. Continuation of the Cycle: The series will likely continue to show the same pattern of gradual increase and
    decrease in brightness. After reaching the latest minimum value, it will start to increase again, reaching
    another peak, and then decrease to another trough.
2. Amplitude and Frequency: The amplitude of the variations will likely remain consistent with the observed
    pattern, with the brightness increasing to a peak similar to the previous maximum (around 2.80) and then
    decreasing to a similar minimum (around -0.57).
3. Possible Variations: While the overall pattern is expected to continue, there may be small variations or
    irregularities due to factors such as:
    - Noise: Random fluctuations that can occur due to observational errors or intrinsic variability.
    - Long-term Trends: Gradual changes in the star's behavior over time, which might cause the amplitude or period
```

```
            to change slightly.
    - External Factors: Events like stellar flares or interactions with a companion star could introduce anomalies.

Visualization:
To better understand the continuation, a visual representation of the time series would show a wave-like pattern,
    with the latest part of the series starting to increase in brightness after reaching a minimum. The next
    cycle would follow a similar trajectory, peaking at a similar value and then decreasing again.

Conclusion:
The time series is likely to continue with a periodic pattern of increasing and decreasing brightness, maintaining
    a similar amplitude and period as observed. Any deviations from this pattern would be interesting and could
    indicate changes in the star's behavior, which would be valuable for further astrophysical study.
<|end_of_text|>
```

# D. Broader Impacts

Our work introduces an architecture leveraging established open-source model components to achieve new capabilities in processing and generating mixed-modal time series and text. While we build upon models that have undergone community vetting, our architecture's application to diverse domains, including healthcare and finance, warrants a careful consideration of broader impacts.

Positively, this approach offers significant potential. By integrating textual context with time series analysis, it can enhance the interpretability and utility of complex data, for example, by aiding in the summarization of ECG signals or providing textual explanations for environmental data trends. This could improve decision-making and accessibility across scientific and industrial applications. The ability to generate multimodal data also holds promise for data augmentation and system testing.

However, we recognize that the combination of text and time series generation, even using existing components, presents specific risks. Our primary concern is the risk of hallucinations if this line of work is used for safety-critical problems. Progress on significantly reducing hallucinations in LLMs, as well as the development of robust safeguards, are important areas for future work, and necessary before this line of work can be used in safety-critical domains. Furthermore, as with any model trained on public data, there is a risk of perpetuating inherent biases if not carefully managed, particularly in sensitive applications like healthcare. Errors in critical tasks, though a risk in any system, could also have significant consequences.

Our research focuses on fine-tuning existing models rather than releasing a new frontier model. Nevertheless, we believe any future deployment of systems based on these methods should incorporate robust safeguards. These would include careful dataset curation to mitigate bias, human-in-the-loop systems for critical decision-making (especially in healthcare), and ongoing research into detecting AI-generated multimodal content. We encourage responsible development practices as this line of research progresses.

From an ethical and societal perspective, TsLLM can democratize powerful modeling tools across healthcare, environmental sciences, and industry, especially in resource-limited settings. However, risks include potential misuse and overreliance in critical decision-making. Future work should improve interpretability, robustness to distributional shifts, and domain-specific validation to ensure responsible use.

