# OpenReview forum: "TsLLM: Augmenting LLMs for General Time Series Understanding and Prediction"
_ICML.cc/2026/Conference — ICML 2026 regular_

### Official Review · Reviewer_JEp2 · 2026-02-23

**Soundness:** 2
**Presentation:** 2
**Significance:** 3
**Originality:** 2
**Overall Recommendation:** 4
**Confidence:** 2

**Summary:**

The authors propose TsLLM, a multimodal generative model that augments a pretrained decoder-only LLM with time series perception and generation abilities. The key idea is to reformulate time series tasks, including forecasting, classification, anomaly detection, and general question-answering, as autoregressive generation problems over interleaved sequences of text and time series tokens. On ECG recordings, questions, and answers across several question types, TsLLM yields consistent accuracy improvements and notable answering quality.

**Compliance With Llm Reviewing Policy:**

Affirmed.

**Ethical Review Concerns:**

N/A.

**Final Justification:**

"I have increased my score as the authors addressed some of my requests in the rebuttal. That said, without seeing the final revision, I'm not sure if the presentation is fully polished, hence the low confidence.

**Key Questions For Authors:**

1. See Weaknesses.
2. The visualization in Figure 2 is quite misleading. To my knowledge, the sequences are input into the large model as numerical data rather than images. Could the authors clarify this format? This is because existing literature [8] has explored using image and text modalities for time series analysis.

### References
[8] Dong, Sixun, et al. "Teaching Time Series to See and Speak: Forecasting with Aligned Visual and Textual Perspectives." *arXiv preprint arXiv:2506.24124* (2025).

**Limitations:**

Yes.

**Strengths And Weaknesses:**

### I. Strengths

1. Despite the fact that research exploring LLM-augmented time series analysis has been prevalent for a while, a unified foundation model is still a timely problem.
2. The fundation model architecture and training strategy are written clearly.
3. Extensive ablation experiments demonstrate the effectiveness of each module.

### II. Weaknesses

1. **Motivation:** Although the authors emphasize the necessity of unifying time series and language models, several aspects of their core motivation are debatable. First, the authors' claim that current research "limits the amount of signal and context that fits in context windows and hinders pattern recognition" is overstated. In fact, since this topic emerged, numerous studies [1,2,3,4,5] have employed encoders to extract features from time series, significantly compressing the number of tokens. Second, unifying all tasks as "next-token prediction" is not unprecedented [1,6], and whether this simplification leads to suboptimal solutions warrants further investigation.
2. **Proposal:** First, the overall TsLLM framework appears to be a simple stacking of existing LLM4TS techniques. Specifically, the interleaving of Time Series Encoding and Decoding with text is similar to ChatTS [1], the cross-attention Adapter corresponds to TimeLLM [3], the introduction of special token markers resembles ChatTS [1] and ChatTime [2] (also Scale-Shape Decomposition), and furthermore, autoregressive generation is likewise nothing novel [6]. Importantly, the connections between them aren't strong enough, or to put it another way, the author hasn't told a coherent story.
3. **Experiment:** I believe there are issues with the experimental focus of the paper. According to the Introduction, the focus should be on general tasks (such as prediction and anomaly detection), but the experiments in the main body are almost entirely Q&A experiments on a single dataset (ECG-QA), and the comparison methods, apart from ChatTime and ChatTs, are mostly non-fine-tuned models or simple methods (such as Linear Regression). I suggest the authors appropriately swap the experiments in the Appendix with the ablation studies in the main text; I would prefer to see experimental results like those in Table 7. Furthermore, from Table 7, it can be seen that TsLLM's leading margin is not particularly prominent, and dedicated large model methods such as TimeLLM and CALF have not been taken into consideration. Additionally, could the datasets be described in a separate section? They appear very confusing across different tasks and scenarios throughout the paper. Alternatively, I recommend using a fixed set of datasets uniformly across all tasks, rather than the highly imbalanced experimental setup currently presented. Although the ablation experiments in the paper are comprehensive, they lack a crucial ablation study that removes the large model backbone, as recent research has questioned the effectiveness of this design [7].
4. **Reproductivity:** This research is application-focused, but the codebase and built dataset is not available.

### References
[1] Xie, Zhe, et al. "ChatTS: Aligning Time Series with LLMs via Synthetic Data for Enhanced Understanding and Reasoning." *Proceedings of the VLDB Endowment* 18.8 (2025): 2385-2398.
[2] Wang, Chengsen, et al. "Chattime: A unified multimodal time series foundation model bridging numerical and textual data." *Proceedings of the AAAI Conference on Artificial Intelligence*. Vol. 39. No. 12. 2025.
[3] Jin, Ming, et al. "Time-llm: Time series forecasting by reprogramming large language models." *arXiv preprint arXiv:2310.01728* (2023).
[4] Zhang, Haochuan, et al. "Tempogpt: Enhancing time series reasoning via quantizing embedding." *arXiv preprint arXiv:2501.07335* (2025).
[5] Liu, Peiyuan, et al. "Calf: Aligning llms for time series forecasting via cross-modal fine-tuning." *Proceedings of the AAAI Conference on Artificial Intelligence*. Vol. 39. No. 18. 2025.
[6] Liu, Yong, et al. "Autotimes: Autoregressive time series forecasters via large language models." *Advances in Neural Information Processing Systems* 37 (2024): 122154-122184.
[7] Tan, Mingtian, et al. "Are language models actually useful for time series forecasting?." *Advances in Neural Information Processing Systems* 37 (2024): 60162-60191.

---

> ### Author Rebuttal · Authors · 2026-03-31
>
> We thank reviewer for the detailed feedback and for acknowledging the timeliness of the problem, the clarity of the architecture description, and the effectiveness of the ablation experiments. We address each concern below.
>
> **Motivation (W1)**
>
> The quoted passage refers specifically to text-only LLMs processing time series as strings, not to time series models or encoder-based LLM methods. The point is that “text tokenization represents each numeric value with multiple tokens, inflating sequence length and obscuring temporal structure.” This limitation of text-only LLMs is well documented (Spathis & Kawsar, 2024; Merrill et al., 2024), and encoder-based methods, including TsLLM, are meant to address it. We will revise the wording to make this explicit.
>
> On next-token prediction, we do not claim this unification is unprecedented. Our claim is that it works effectively at scale for interleaved text and time series across diverse tasks, and we will temper any language suggesting novelty for autoregression itself.
>
> **Relationship to prior work (W2)**
>
> We respectfully disagree with the characterization that TsLLM is a “simple stacking” of existing techniques. While the ingredients have precedents, the specific design choices and their interactions required substantial experimentation and produced non-obvious outcomes.
>
> Our encoder-decoder uses a beta-VAE with a carefully tuned annealing schedule. Table 5 shows that alternatives such as VQ-VAE, different beta values, and different patch sizes lead to substantially worse downstream performance, and that the relation between reconstruction quality and downstream performance is non-monotonic.
>
> Our cross-attention adapter is initialized from sampled LLM embeddings, constraining time series tokens to lie near the LLM’s embedding manifold; Table 4 shows this outperforms MLP and linear alternatives.
>
> The three-stage training pipeline was adopted because early experiments with all parameters unfrozen from the start were unstable, and the data pipeline produces 25B+ tokens across 20+ datasets and multiple task types.
>
> We believe this coherent integration, supported by ablations showing that these choices matter, is a meaningful systems-and-methodology contribution.
>
> **Experimental focus and baselines (W3)**
>
> The main body focuses on cross-modal tasks because these are TsLLM’s primary target: capabilities requiring both temporal signals and language understanding. It contains three evaluations: ECG-QA (Table 1), TimeSeriesExam (Table 2), and Context Is Key (Table 3). Traditional time series tasks are in the appendix because they are not TsLLM’s main target; TsLLM’s competitive (but not leading) performance on them is included for completeness.
>
> We prefer to keep the ablations in the main text because they provide the clearest insight into why TsLLM works, especially the reconstruction–performance tradeoff (Table 5), the adapter comparison (Table 4), and scaling with LLM size.
>
> For Table 7 (GIFT-Eval forecasting), methods such as Time-LLM and CALF are not foundation models. They require dataset-specific training for each of the 23 datasets, whereas all baselines in our comparison are zero-shot or few-shot. Including Time-LLM would therefore require 23 separate training runs and compare fundamentally different evaluation paradigms.
>
> For Table 3 (CiK), traditional TSFMs cannot use the textual context that defines the benchmark. Including them would only show that models without relevant context perform worse, which is expected. We agree this baseline choice should be explained more explicitly.
> On the suggested ablation removing the LLM backbone: work such as Tan et al. (2024) questions LLM backbones for purely numerical time series tasks where language capabilities are never exercised. That is valid in that setting. However, TsLLM’s core tasks(QA, contextual forecasting, and report generation) fundamentally require language understanding and generation. Removing the LLM backbone would remove the ability to perform these tasks at all. This is categorically different from using an LLM only as a pretrained transformer for numerical prediction.
>
> **Reproducibility (W4)**
>
> We have now uploaded the complete codebase to https://anonymous.4open.science/r/tsllm-icml-anom-A5B0/ .
>
> **Figure 2 visualization (Q1)**
>
> The time series in Figure 2 are shown as plots purely for illustration. The actual inputs to TsLLM are dense vector representations produced by the encoder, not images.
> Image-based approaches (Daswani et al., 2024; Liu et al., 2025) were discussed in related work. TsLLM deliberately avoids this approach in favor of a dedicated encoder that preserves numerical precision.
>
> **Limitations**
> We did include a limitations discussion in Section 6, covering computational cost, performance on pure numerical tasks, and the lack of standardized cross-modal benchmarks. We would welcome feedback on any specific limitations to expand this discussion.

---

> > ### Author Rebuttal · Reviewer_JEp2 · 2026-04-01
> >
> > I appreciate the authors’ detailed response. First, given that the authors have provided the code and addressed some of my concerns, I have accordingly increased my score. Second, I believe the current presentation of the paper has non-negligible problems; this is not a challenge to the overall contribution of the work.
> > Specifically, the start of Sec. 3 outlines four components of the framework, but these are not clearly elaborated on one by one in the subsequent text. Instead, various design choices (e.g., β-VAE) appear within different subsections without sufficient motivation or insight, leaving the reader insufficiently guided (hence my comment that the paper reads like “a simple stacking of existing LLM4TS techniques.”).
> > In addition, regarding the ablation question about not being able to remove the LLM due to its NLP capability, I would argue that much of the corpus used to pretrain LLMs is not actually relevant to time-series tasks. Why not use a much smaller, non-pretrained LM (Transformer or Bi-LSTM) and train it end-to-end instead?

---

> > > ### Author Response · Authors · 2026-04-07
> > >
> > > Thank you for the careful review of the paper, rebuttal follow-up, and specific suggestions. We have revised Section 3 to better connect the model components, design goals, and specific design choices, and we ran an additional backbone ablation to directly test replacing the pretrained LLM with smaller, non-pretrained LMs.
> > >
> > > - You make a good point about the presentation of the methods section and how we didn’t connect the subsections and the design choices back to the core list of components and the motivations/design goals. In the revision, we have reorganized Section 3 so that the four components introduced at the start of the section are presented explicitly and in the same order. We will also highlight those components more clearly in Figure 1. We revised the text to explain the motivations behind each of the design choices more clearly and in direct connection to those components and design goals. For instance, the use of the β-VAE encoder-decoder as a way to pretrain compact, regularized time-series representations before alignment with the LLM is now better motivated.
> > > - While it is true that much of the corpus used to pretrain LLMs is not directly applicable to time series analysis, we have found that LLM pretraining is useful for TsLLM. The general and domain-specific knowledge, language capabilities, and reasoning abilities of pretrained LLMs can be valuable for understanding context and instructions, and cannot be learned in the approximately 15B tokens of text that TsLLM is trained on. To empirically validate this, we trained otherwise identical TsLLM models using the same architecture, dataset, and training procedure, but replaced the pretrained backbone with randomly initialized language models, namely, GPT2-small (124M params transformer) and SmolLM2-360M (with random initialization, matching setup of the SmolLM2-360M baseline reported in Table 4). The non-pretrained GPT2-small-based TsLLM variant achieves 43.5% accuracy on ECG-QA and the non-pretrained SmolLM2-360M variant achieves 43.7% accuracy, whereas TsLLM with a pretrained SmolLM2-360M backbone achieves 59.6% accuracy. This result implies that general language model pretraining provides substantial benefit beyond model capacity alone in our setting. We will add this ablation study to the revised paper in Section 5; thank you for the suggestion.

---

### Official Review · Reviewer_56kc · 2026-03-03

**Soundness:** 3
**Presentation:** 3
**Significance:** 3
**Originality:** 3
**Overall Recommendation:** 4
**Confidence:** 5

**Summary:**

TsLLM bridges a key gap between LLMs and time-series models: conventional methods handle numeric forecasting well but lack contextual and language capabilities, while LLMs struggle with raw time-series representations. It augments a pretrained LLM with a patch-based time-series encoder–decoder and trains on 25B+ tokens of interleaved time-series–text data across forecasting, QA, anomaly detection, classification, and report generation via next-token prediction. TsLLM demonstrates strong performance and zero-/few-shot generalization on tasks requiring joint time-series and language understanding.

**Compliance With Llm Reviewing Policy:**

Affirmed.

**Final Justification:**

I have read the rebuttal and will keep my positive score.

**Key Questions For Authors:**

1. You mention a β-VAE objective for pretraining the encoder-decoder. Could you elaborate on the design of this objective? What value of β did you use, and how was it selected?
2. The 25 billion token training corpus is impressive. Could you provide more details about the composition of this corpus? What proportion is real vs synthetic data and text vs time series? How were the different tasks weighted during training?
3. You demonstrate strong few-shot performance. How were the few-shot examples selected? Did you observe significant variation in performance based on example selection?

**Limitations:**

1. The paper evaluates primarily on a narrow set of cross-modal benchmarks (ECG-QA, TimeSeriesExam, CiK).

**Strengths And Weaknesses:**

S1: The paper tackles an important problem where experts need to analyze time series alongside contextual information.
S2: The scale of the training data (over 25 billion tokens) is impressive and likely crucial to the model's strong performance. The diversity of tasks included in training is also notable.
S3: The paper evaluates TsLLM on a wide range of tasks including time series QA, ECG QA, contextual forecasting, traditional forecasting, and classification.

W1: It's not clear how well TsLLM handles general multivariate time series, especially with many channels.
W2: For some tasks, the exact evaluation protocols are not entirely clear. For example, how were the few-shot examples selected? What exactly constitutes "contextual forecasting" versus standard forecasting?

---

> ### Author Rebuttal · Authors · 2026-03-31
>
> We would like to thank the reviewer for the thorough review and for recognizing the importance of the problem, the scale of our training data, and the breadth of our evaluation. We address each point below.
>
> **Handling of multivariate time series (W1)**
>
> TsLLM handles multivariate time series by encoding each variate independently and feeding them sequentially into the LLM. This design choice is deliberate. First, it provides maximum flexibility: the model can handle any number of variates without architectural changes, which is essential for a foundation model that must generalize across domains with very different dimensionality. Second, it avoids baking inter-variate relationships into the encoder that may not transfer across domains. Third, this is consistent with how LLMs handle multi-document or multi-image inputs in other modalities: the model learns cross-variate relationships through attention over the full context, rather than through explicit structural assumptions in the encoder.
>
> In practice, TsLLM works well on moderately multivariate data. ECG-QA uses 12-lead ECGs (12 variates), and the GIFT-Eval multivariate results in Table 7 show competitive performance. For highly multivariate signals (hundreds of channels), we do not yet have reliable benchmarks or sufficient training data to make strong claims, and the primary constraint would be context window length, since each additional variate adds tokens to the sequence. This is worth exploring in future work, potentially using synthetic data to construct controlled experiments. We will add a more explicit discussion of this tradeoff in the revised paper, and make it more explicit in Section 6.
>
> **Evaluation protocol clarity (W2)**
>
> We thank the reviewer for highlighting this lack of clarity and will add more detailed protocol descriptions in the revision.
> Contextual forecasting refers to forecasting tasks where textual context provides essential information for accurate prediction that is not available from the numerical history alone. This definition follows the Context Is Key (CiK) benchmark (Williams et al., 2024). The key distinction from standard forecasting is that the textual context is necessary for strong performance, not merely supplementary.
> For few-shot example selection: in classification tasks with fewer than 10 classes, we sampled 1 to 3 question-time series-answer triplets uniformly at random for each class, ensuring equal class representation. For tasks with 10 or more classes, examples were sampled without class stratification to keep the prompt within context length limits. For ECG-QA, few-shot examples were drawn from the same question type. We did not run a systematic study of sensitivity to example selection, though preliminary experiments suggested moderate but not dramatic variation. We will add this protocol explicitly to the appendix so the evaluation is fully reproducible.
>
> **Beta-VAE details (Q1)**
>
> We used a sigmoid-based annealing schedule for beta, starting at approximately 0.5 and increasing to a peak of 1.25 over the course of training. This was found empirically to give good convergence and strong reconstruction on a held-out validation subset of the GIFT-Eval Pretrain dataset. Starting with a low beta prioritizes reconstruction accuracy early on, while the gradual increase encourages smoother and more structured latent representations. The ablation in Table 5 shows that both lower and higher beta values lead to worse downstream performance, confirming that getting this balance right matters. We will add these details to the paper.
> Training corpus composition (Q2)
>
> The approximate breakdown of our ~27B token training corpus is as follows. About 12B tokens come from real-world datasets and about 15B from synthetic data generation. In terms of modality, roughly 15.5B are text tokens and 11.3B are time series tokens. The task breakdown is approximately 16.2B tokens for QA, 8.8B for forecasting, 1.8B for classification, and under 1B for other tasks (anomaly detection, segmentation, report generation). This distribution reflects the natural composition of our collected and synthesized data; we did not apply explicit task re-weighting during training. A detailed per-dataset breakdown is in Table 9 (Appendix B).
> Breadth of cross-modal benchmarks
>
> We have discussed this as a limitation of the field in Section 6. ECG-QA, TimeSeriesExam, and CiK are, to our knowledge, the most established benchmarks that test joint time series and language understanding. We supplement these with traditional benchmarks (GIFT-Eval, UCR-TSC, MSL) and text-only evaluations (MMLU-Pro, LiveBench, GPQA, MATH, GSM8K) to be as comprehensive as possible. We hope that the release of our code and data will help push forward the development of additional cross-modal benchmarks.

---

> > ### Author Rebuttal · Reviewer_56kc · 2026-04-03
> >
> > Thanks for the rebuttal, I will keep my score, which is already positive.

---

### Official Review · Reviewer_r9tt · 2026-03-13

**Soundness:** 3
**Presentation:** 3
**Significance:** 3
**Originality:** 3
**Overall Recommendation:** 5
**Confidence:** 3

**Summary:**

This paper compresses time series into patch embeddings, injects them into an LLM via an adapter, and trains everything uniformly under a next-token prediction objective, enabling the LLM to understand both time series and text. This addresses two structural limitations: traditional time series models lack language understanding, while vanilla LLMs handle numerical signals inefficiently and imprecisely through text tokenization. Concretely, the paper proposes a patch-based β-VAE encoder-decoder to compress temporal signals into compact representations, a cross-attention adapter to inject these representations into the pretrained LLM's token stream, and a unified next-token prediction framework that covers all task types, with a three-stage training procedure to complete modality alignment and capability activation.

**Compliance With Llm Reviewing Policy:**

Affirmed.

**Final Justification:**

I have read the rebuttal and the "originality" scores have improved from 2:fair to 3:good.

**Key Questions For Authors:**

See weaknesses above.

**Limitations:**

Yes

**Strengths And Weaknesses:**

The problem motivation is well-grounded. The gap between traditional time series models and pure LLMs is real and practically significant. But at its core, the paper is essentially an A+B integration work. Its strengths lie in the fact that the core design choices are well-justified and the unified next-token prediction framework is elegant. In addition, the ablation studies are thorough, covering backbone size, adapter architecture, encoder type, and encoder hyperparameters, and they yield non-trivial insights such as the trade-off between reconstruction fidelity and downstream performance.

In terms of weaknesses,
First, the baseline selection is overly conservative. Comparing against vanilla LLMs on contextual forecasting is not a strong test, since those models are structurally disadvantaged at processing numerical inputs. Why not consider some more competitive baselines such as TSFM or Time-LLM? I think this would make the empirical claims more convincing.

Second, the heavy reliance on synthetic data is underexplored. Roughly 15 out of 27 billion training tokens are synthetically generated, yet there is no systematic analysis of how this affects generalization or whether the model overfits to synthetic distributions.

---

> ### Author Rebuttal · Authors · 2026-03-31
>
> We thank the reviewer for the positive assessment and for recognizing the well-grounded motivation, justified design choices, and thorough ablation studies. We address the two identified weaknesses below.
>
> **Baseline selection for contextual forecasting (W1)**
>
> We appreciate this concern. Our baseline selection for the CiK benchmark was guided by the types of models that can actually perform contextual forecasting, meaning forecasting that conditions on both numerical history and unstructured text. Traditional TSFMs and methods like Time-LLM operate exclusively on numerical inputs and cannot incorporate the textual context that is central to the CiK benchmark’s design. Including them would require ignoring the textual context entirely, which would not be a meaningful comparison for this task.
>
> That said, we agree the comparison could be strengthened. We will add results for TSFMs run in their standard (context-free) mode as an additional reference point, making it explicit that these models cannot use the textual context. This will help readers understand the gap between context-free and context-aware forecasting, which is exactly the capability TsLLM targets. We will also clarify this framing in the paper so that the CiK evaluation is understood primarily as a test of contextual point forecasting, not as a claim that TsLLM is the strongest forecaster in every setting.
>
> For the traditional forecasting results in Table 7 (GIFT-Eval), we do compare against strong TSFM baselines including Moirai, Chronos, and TimesFM. We did not include methods like Time-LLM or CALF because they are not foundation models and would need to be retrained from scratch for each of the 23 datasets in GIFT-Eval, unlike all the zero/few-shot baselines in our comparison. We will clarify this distinction in the revised paper.
>
> **Synthetic data analysis (W2)**
>
> We thank the reviewer for this fair and important point. To address it, we have trained three smaller-scale variants of TsLLM (all using Qwen2.5-3B, trained for 10B tokens) to isolate the effect of synthetic data:
> •	Variant A (real data only): trained exclusively on curated real-world datasets.
> •	Variant B (synthetic data only): trained exclusively on our synthetic QA and task data.
> •	Variant C (mixed): trained on a 50/50 mixture of real and synthetic data.
>
> An important nuance in interpreting these results is that the real and synthetic portions of our dataset are not interchangeable in terms of task coverage. The synthetic data is predominantly general QA, while the real data covers forecasting, classification, anomaly detection, and domain-specific tasks like ECG-QA. So differences in performance across tasks reflect not just real-vs-synthetic quality, but also which tasks were represented in training. We will present these results in this context and discuss what they reveal about the role of synthetic data in TsLLM’s training. More broadly, we will revise the paper's wording so that "synthetic" is not treated as a weakness in itself; the relevant issue is whether the synthetic data is sufficiently diverse and well-matched to the target capabilities.
>
> **On the “A+B integration” characterization**
>
> While we understand this characterization, we think it understates the difficulty of making these components work well together at scale. The specific design choices (how to pretrain the encoder-decoder, how to bridge modalities, how to structure multi-stage training, how to curate and synthesize 25B+ tokens of data) each required substantial experimentation, and our ablation studies show that many of these choices matter a lot. Replacing our beta-VAE encoder with a VQ-VAE drops accuracy by 12.1%. Using a linear adapter instead of cross-attention drops it by 6.4%. The reconstruction-vs-downstream-performance tradeoff documented in Table 5 is genuinely non-obvious. We believe this careful integration, validated by ablations showing each decision matters, goes beyond simple combination of existing parts. That said, we agree that the contribution is best framed as a systems-and-methodology contribution rather than as the invention of entirely new building blocks, and we will adjust the paper’s positioning accordingly.

---

> > ### Author Rebuttal · Reviewer_r9tt · 2026-04-04
> >
> > Thank you for the detailed rebuttal. Your response on the 'A+B integration' concern is convincing. I have accordingly raised my originality score.

---

### Official Review · Reviewer_Sqcv · 2026-03-13

**Soundness:** 2
**Presentation:** 2
**Significance:** 3
**Originality:** 2
**Overall Recommendation:** 3
**Confidence:** 4

**Summary:**

This paper proposes TsLLM, a decoder-only LLM augmented with a time-series encoder/decoder and adapter layers so that text and time-series segments can live in a unified autoregressive framework. The main idea is to represent time series through a patch-based latent interface, combine these with a pretrained LLM, and train the whole thing on a large mixture of time-series-only, paired text/time-series, and synthetic instruction-style data. The authors position TsLLM as a general multimodal foundation model for tasks like contextual forecasting, QA over time series, classification, anomaly detection, and report generation. The strongest results are on tasks that need both temporal signals and language, while on standard numerical benchmarks the model is competitive but not state of the art, and it roughly keeps the base LLM's text performance.

**Compliance With Llm Reviewing Policy:**

Affirmed.

**Final Justification:**

I would increase my acknowledgement to (b) Partially resolved and keep my score unchanged at this stage.

The rebuttal and follow-up are constructive: the authors corrected the CiK metric statement, acknowledged non-comparability to the official CiK protocol, toned down novelty framing, expanded related-work positioning, and released clearer experiment configs (including an ECG-QA/PTB-XL- excluded setup). These are meaningful improvements.

However, core confidence issues remain only partially resolved. The empirical claims still lack a fully auditable run-level evidence chain (exact run/config/commit/result linkage, complete logs/artifacts for key rebuttal claims), and the CiK results still cannot support benchmark-level SOTA under the official protocol. The new clarifications improve trust, but do not yet fully close reproducibility and evaluation-validity gaps.

**Key Questions For Authors:**

1. I need to understand the exact train/test isolation for ECG-QA. Did TsLLM's training mixture include only the official ECG-QA training split, or were validation or test examples also present? My interpretation of Table 1 changes a lot depending on the answer.

2. Can the authors run TsLLM on Context Is Key using the benchmark's official RCRPS protocol and compare directly against the reported strongest baselines, especially the Direct Prompt baselines? This would tell me a lot about whether the contextual forecasting claim actually holds.

3. Why are Multi-Modal Forecaster and TimeText Corpus not discussed in the main related work, given how close they are to what TsLLM is doing? If the authors think these are fundamentally different, I want to hear why.

4. I would like to see a decomposition of what the architecture contributes versus what the large training mixture contributes. For example, what happens with the same mixture but a weaker time-series interface, or with the current architecture trained on a smaller mixture comparable to what prior multimodal time-series LLMs used? Without this I cannot tell if the contribution is the model or the data.

5. Was any code, preprocessing logic, or implementation snapshot provided at submission time? If not, I think this is a significant problem given how much the paper's contribution depends on data curation and pipeline engineering.

**Limitations:**

Partially. The paper discusses computational cost, benchmark immaturity, and the fact that the model is not SOTA on pure numerical tasks. But it does not discuss the reproducibility problem, it does not acknowledge that training on ECG-QA while using ECG-QA as a key headline benchmark is a conflict that needs careful handling, and it does not discuss how sensitive the results might be to data-mixture and evaluation-protocol choices. I think these are the limitations that actually matter.

**Strengths And Weaknesses:**

Strengths:

1. I think a model that can process both numerical time series and natural language in one framework is a genuinely useful thing to have, and the observability gap between text-only LLMs and numerical TSFMs is real. The paper is going after a problem worth solving.

2. I find the overall systems design to be coherent. The individual ingredients — patch-based latent representation, scale handling, adapter fusion, autoregressive formulation across tasks — are not new by themselves, but I think putting them together at this scale is real work and the design makes sense to me as an engineering effort.

3. The reported results on the paper's preferred multimodal tasks are meaningfully positive. The gains over text-only LLM baselines and prior time-series LLM baselines on ECG-QA, TimeSeriesExam, and contextual forecasting suggests that the approach is doing something nontrivial rather than just riding on the base LLM.

4. I appreciate the ablations, especially the negative finding that optimizing reconstruction quality alone does not lead to good downstream performance. The text retention results are also valuable — I would have been very concerned if the base LLM degraded badly after all this training.

Weaknesses:

1. I think the novelty is overstated by a good amount. The paper repeatedly frames its contribution as enabling capabilities that prior work cannot provide, and I don't buy this. ChatTime already explicitly claims bimodal text/time-series input-output, contextual forecasting, and time-series QA. Multi-Modal Forecaster also studies joint text and time-series forecasting, and TimeCAP/ITFormer cover neighboring slices of the capability space. The novelty does not live where the paper says it does — the real contribution is a stronger and more unified large-scale instantiation of multimodal time-series modeling, not first-of-its-kind capabilities. I think the paper would be more convincing if it framed itself honestly rather than overclaiming.

2. The related work section is incomplete in exactly the places where the novelty case is most fragile. ChatTime is described too narrowly, leaving out its own claim of unified multimodal modeling and bimodal input/output. Multi-Modal Forecaster shows up in the reference list but is not discussed in the main related work, despite being one of the closest neighbors to TsLLM conceptually. Time-MMD is mentioned later in the appendix, but its relevance to multimodal contextual forecasting is not properly brought into the novelty discussion. I feel like this makes it hard for me to judge what exactly is new.

3. The ECG-QA headline result is not a fair comparison free of contamination. The paper does disclose that TsLLM was trained on ECG-QA, and says this is only a small fraction of total training tokens, but the absolute amount is still large. Meanwhile several baselines in the main table were not trained on ECG-QA at all and only received few-shot exemplars at inference time. I don't think the result is wrong per se, but the paper should not be using this table as strong evidence of few-shot or out-of-domain generalization. I would call this an in-domain multitask result unless the authors can show me exact train/test isolation.

4. I cannot make sense of the Context Is Key evaluation relative to the benchmark's intended setup. The paper reports SMAPE/MASE, but the CiK benchmark paper centers evaluation around RCRPS and compares against strong Direct Prompt baselines. Because TsLLM is not evaluated under the benchmark's own metric and does not directly compare against the benchmark's strongest baselines under that metric, I don't think the paper establishes what it claims to establish. I see this as pointing in a positive direction, but it is not a clean CiK victory.

5. I think the TimeSeriesExam result needs to be taken with some caution. TimeSeriesExam is synthetic, and TsLLM's training mixture contains a very large amount of synthetic QA-style data. I don't think there is leakage, but I do think the result is closer to in-distribution validation than to evidence that the model generalizes well to real-world scenarios.

6. The paper criticizes prior work for relying on synthetic data, but TsLLM is not innocent of the same thing. I understand the criticism is aimed at prior work's relatively small and sometimes entirely synthetic training sets, while TsLLM uses a much larger mixed corpus. But the paper's wording gives the impression that synthetic data itself is the core problem, even though TsLLM also relies on it heavily. I think the distinction the paper is actually making is about scale and diversity, not about "synthetic" as such, and the writing should reflect that.

7. Reproducibility is where I lose the most confidence. For a paper whose contribution depends this heavily on data curation, training mixture design, task formatting, and evaluation protocol, I need to be able to inspect the implementation. No code or anonymized implementation appears to be available. Without this I cannot tell which gains come from the architecture, which from the data mixture, and which from benchmark-specific engineering choices. This matters much more here than it would for a paper proposing a single clean algorithm.

Overall assessment by dimension:

- Soundness: I don't see an obvious fatal flaw in the method itself, and I believe the positive results are probably genuine. But several central claims are not supported as tightly as they need to be, mainly because of evaluation fairness and reproducibility issues.
- Presentation: the paper reads well at the high level, but the related work discussion is not honest enough about what already exists, and important experimental details are missing from places where they would matter.
- Significance: the problem is important, and if the claims held up better I think this line of work could have real impact. Even as it stands the paper might push the field forward.
- Originality: there is novelty, but it lives in system integration and scaling, not in first-of-its-kind capabilities. The paper currently oversells the originality and this hurts more than it helps.

---

> ### Author Rebuttal · Authors · 2026-03-31
>
> We thank the reviewer for the careful and constructive evaluation.
>
> **Novelty claims and related work (W1, W2)**
> We regret the imprecise wording in the abstract and introduction. When we wrote that TsLLM enables "capabilities that existing approaches cannot provide," we were referring to traditional time series models and text-only LLMs, not to all existing work. We fully acknowledge that recent methods like ChatTime and TimeCAP have also aimed to unify text and time series modeling. Our related work discusses ChatTS and TimeCAP, but the treatment of ChatTime and Multi-Modal Forecaster was insufficient.
>
> We will revise the abstract and introduction to remove any first-of-its-kind language and expand the related work accordingly. TsLLM's contribution lies in (1) a high-fidelity pretrained encoder-decoder avoiding precision loss from quantization or image-based approaches, (2) full interleaved input-output support enabling a broader task range, (3) training at significantly larger scale (25B+ tokens), and (4) the resulting empirical gains across diverse benchmarks. We will make sure the writing reflects this honestly.
>
> **ECG-QA evaluation fairness (W3, Q1)**
> TsLLM was trained only on the official ECG-QA training split; no validation or test examples were included. But we agree that even training-split exposure creates an asymmetry relative to zero/few-shot baselines.
>
> We have trained a variant that completely excludes both ECG-QA and PTB-XL from its training data. This variant achieves accuracies of 68.5/50.6/32.7 (S-Verify/S-Choose/S-Query), averaging 50.8 – substantially better than all other foundation models without ECG-QA training, while falling below specialized fine-tuned models. This is consistent with our claims: TsLLM does not aim to outperform specialized models on their target datasets, but significantly outperforms general-purpose models and demonstrates strong cross-domain transfer. We will add this variant to Table 1.
>
> **Context Is Key evaluation protocol (W4, Q2)**
> TsLLM's decoder produces deterministic patch reconstructions and cannot currently generate distributional forecasts, so we cannot compute the RCRPS metric used as CiK's primary evaluation. We acknowledge this limitation and will discuss it explicitly.
>
> That said, SMAPE and MASE are standard forecasting metrics also reported in the CiK paper. TsLLM's substantial margins over both traditional ML methods and frontier LLMs on these metrics show it can effectively integrate textual context with numerical forecasting. We will note the RCRPS absence transparently, discuss probabilistic output support as future work, and avoid implying direct comparability with methods evaluated under the full probabilistic protocol.
>
> **TimeSeriesExam and synthetic data concerns (W5, W6)**
> We will add nuanced discussion of the TimeSeriesExam result. Two clarifications: First, our synthetic QA pipeline uses different templates, time series sources, and generation models from TimeSeriesExam's construction—the overlap is in task type, not data distribution. Second, a foundation model's ability to handle a broad range of tasks becoming in-distribution is a strength, especially given the large gap over frontier models.
>
> We agree the relevant distinction in prior work is about scale and diversity, not whether data is synthetic per se. TsLLM relies heavily on synthetic data, as do most methods given the scarcity of paired time series-text datasets. Our data contribution is in the scale and diversity of synthesis. We will revise the TimeSeriesExam discussion to present it as a synthetic stress test rather than strong evidence of real-world OOD generalization.
>
> **Reproducibility (W7, Q5)**
> The complete codebase is now at https://anonymous.4open.science/r/tsllm-icml-anom-A5B0/. The dataset can be fully reproduced from released code and public source data.
>
> **Architecture vs. data mixture decomposition (Q4)**
> The ablation in Table 4 provides partial evidence: varying adapter architecture and encoder type while holding data constant shows architectural choices matter substantially (e.g., VQ-VAE drops accuracy by 12.1%). Our new ECG-QA-excluded variant shows the model retains strong QA performance without task-specific training data, suggesting real cross-domain transfer. The gap to the full model confirms task-specific data also helps.
>
> A full factorial decomposition is computationally prohibitive at scale (800 H100-hours per run). We have conducted additional smaller-scale experiments (described in our response to Reviewer r9tt) and will add a more thorough discussion, being careful not to overclaim clean causal separation between architecture and data-mixture effects.
>
> **Limitations discussion**
> We will expand the limitations section to address: (1) the reproducibility gap at submission (now resolved), (2) ECG-QA training overlap and its interpretive implications, (3) inability to produce probabilistic forecasts, and (4) sensitivity to data mixture choices.

---

> > ### Author Rebuttal · Reviewer_Sqcv · 2026-04-04
> >
> > Thank you for the detailed and constructive rebuttal, and for releasing an anonymized codebase. I appreciate (i) the commitment to revise the abstract/intro to remove any first-of-its-kind language and to expand the related-work discussion (especially for ChatTime and Multi-Modal Forecaster), and (ii) the clarification that ECG-QA training used only the official training split.
> >
> > However, after a brief audit of the released repository, I still have material concerns about evaluation validity and reproducibility that are not yet resolved by the rebuttal:
> >
> > - I could not find a config/script/log/checkpoint for the “ECG-QA and PTB-XL excluded” variant reported in the rebuttal, so the new 50.8 average accuracy cannot currently be independently verified.
> > - The repository appears to include a cached-dataset pathway that can set ECG-QA to load all splits (train/val/test) when building the cache; please clarify whether any of the reported models/tables were trained from such cached data, and if not, point to the exact training config/commit used for the main results.
> > - The code does not clearly enforce the claimed GIFT-Eval pretrain/eval separation (the split helper looks effectively inactive), so the “no test leakage” claim remains hard to verify from the release.
> > - For CiK, the evaluation still does not follow the benchmark’s intended protocol (RCRPS/constraint-satisfaction), and the rebuttal statement that CiK “also reports SMAPE/MASE” is factually incorrect; this should be corrected, and the paper should avoid implying benchmark-level comparability under the official metric.
> >
> > Overall, the rebuttal improves the paper’s positioning and partially addresses fairness concerns, but the remaining issues above still substantially limit my confidence in the key empirical claims unless clarified with exact configs/commits and verifiable artifacts.

---

> > > ### Author Response · Authors · 2026-04-07
> > >
> > > Thank you for the follow-up and for taking the time to inspect the released repository. We understand that these issues affect confidence in the empirical claims, and we address each point directly below.
> > >
> > > - In order to make the experiments and results easily reproducible we have refreshed the config files in the shared repository under `tsllm/configs/experiments/` and organized them by experiment. These were converted from the configs saved to our experiment tracker (weights & biases), so they will match what was actually run better than the default configs that were previously included. The new “ECG-QA and PTB-XL excluded” experiment configs you mentioned are now in `tsllm/configs/experiments/rebuttal-ecgqa-excluded`.
> > > - The cached dataset pathway allowing all splits to be used when building the cache for ECG-QA is a remnant of an old experiment, and was not used in any of the experiments reported in the paper (nor was the cached dataset pathway itself). During early development (over a year ago), we observed unexpectedly poor validation performance despite seemingly normal training performance. To determine whether this was due to a bug or if the model was overfitting, we ran a diagnostic experiment in which we trained on the val+test split and evaluated on that same data. Evaluation performance remained poor, which pointed to an issue rather than overfitting, and we traced this to a bug in the evaluation data processor that dropped the time-series sequences. This option was not used after that experiment, and we actually stopped using the cached dataset pathway altogether as we made performance improvements to the dataloading system, but we never actually removed the code. No result reported in the paper or rebuttal used ECG-QA validation or test examples for training. We have updated the repository to remove the dataset caching pathway and other unused code to try to minimize confusion moving forward.
> > > - GIFT-Eval Pretrain (https://huggingface.co/datasets/Salesforce/GiftEvalPretrain) and GIFT-Eval (https://huggingface.co/datasets/Salesforce/GiftEval) are two entirely separate datasets, handled by different dataset loaders. Specifically, GIFT-Eval Pretrain is handled by `GiftTrainingDataset` and GIFT-Eval is handled by `GiftEvalDataset`, which is why they do not internally handle the train/test split – it is already handled by the choice of dataset. In all of our experiments, we strictly maintain the correct train/test split. The exact set of datasets used for training in each experiment can be found in the updated config files.
> > > - We previously stated in the rebuttal that CiK also reports SMAPE/MASE; that was incorrect and we apologize for the error. We agree that our CiK evaluation does not follow the benchmark’s official protocol and is therefore not directly comparable to methods evaluated under the official CiK setup. We will make it very clear in the revised paper that we do not follow the original protocol and explain why (TsLLM doesn’t support uncertain predictions, which are needed to compute the metrics used in the CiK paper). However, we still feel that CiK is a useful benchmark to include because the data and forecasting task is still highly relevant to our work, even if we use different metrics.

---

### Decision · Program_Chairs · 2026-04-30

**Decision:**

Accept (regular)

**Comment:**

This paper presents a large-scale multimodal time-series modeling methodology, TsLLM which comprises of a decoder-only LLM augmented with a time-series encoder/decoder and adapter layers. While the individual components are not novel by themselves, the reviewers appreciated that the design choices are well-justified, the unified next-token prediction framework is elegant, and the ablation studies are thorough and convincing. The scale of the training data (over 25 billion tokens) is  impressive. Overall, this is a solid paper, and the architecture design choices, training data construction, and evaluation methodology in this paper will be quite useful for the multimodal time-series research community.

Please address the weaknesses pointed to by the reviewers in the final version of the paper - these include elaborating more on the synthetic training data analysis, ECG-QA evaluation fairness, and reproducability concerns around the codebase and data